# Iterative Vectors: In-Context Gradient Steering without Backpropagation

Yiting Liu [1]   Zhi-Hong Deng [1]

## Abstract

In-context learning has become a standard approach for utilizing language models. However, selecting and processing suitable demonstration examples can be challenging and time-consuming, especially when dealing with large numbers of them. We propose Iterative Vectors (IVs), a technique that explores activation space to enhance in-context performance by simulating gradient updates during inference. IVs extract and iteratively refine activation-based meta-gradients, applying them during inference without requiring backpropagation at any stage. We evaluate IVs across various tasks using four popular models and observe significant improvements. Our findings suggest that in-context activation steering is a promising direction, opening new avenues for future research.

## 1. Introduction

Few-shot learning has long been a prominent research focus. Recently, language models (LMs) have demonstrated the capability for few-shot learning via in-context learning (ICL) (Brown et al., 2020). In this approach, learning a new task involves conditioning on a few support examples and predicting suitable tokens to complete a query input—all without requiring any parameter updates. This method is appealing because it relies solely on inference, allowing for quick adaptation to various downstream tasks.

However, it has been noted that despite its potential, the predictions of LMs can be highly volatile when conditioned on prompts. The outcomes depend significantly on the templates, demonstrations, and their permutations, and may even ignore or violate the prompt's instructions (Webson & Pavlick, 2022; Min et al., 2022b). Consequently, such variability introduces uncertainty, compromising the reliability

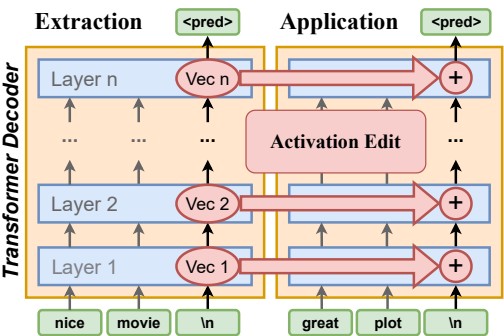

Figure 1. A general illustration of how activation vectors compared in this work improve ICL performance by extracting and editing model activations.

and usability of ICL. Another issue with directly implementing a many-shot ICL approach for LMs is the inherent constraint of context length. This limitation restricts the amount of information that can be provided to the model in a single prompt, potentially hindering its ability to learn effectively from numerous examples. Furthermore, in theory, the inference time increases quadratically as more examples are appended to the query. If examples are lengthy, it may be infeasible to process them within the desired timeframe or model context length.

In this paper, we introduce Iterative Vectors (IVs), a new method for enhancing ICL. As illustrated in Figure 1, rather than staying in the discrete prompt space, IVs delve into the extensive activation space of the model. This exploration reveals a largely uncharted area for developing new methods, with our pioneering efforts to demonstrate how ICL can be enhanced through the model's internal representations.

Iterative Vectors are computed by analyzing the difference between the attention activations of query pairs generated with and without preceding examples during inference. This process aims to capture the insights the model acquires from demonstration examples. These IVs are iteratively reintegrated into the model to produce increasingly stable and effective vectors, progressively incorporating information from additional examples. The refined IVs can subsequently be applied to future inference tasks. This approach preserves the inherent structure of the ICL framework and introduces minimal computational and memory costs, ensuring both

[1]State Key Laboratory of General Artificial Intelligence, School of Intelligence Science and Technology, Peking University. Correspondence to: Zhi-Hong Deng <zhdeng@pku.edu.cn>.

*Proceedings of the 42nd International Conference on Machine Learning*, Vancouver, Canada. PMLR 267, 2025. Copyright 2025 by the author(s).

practical efficiency and methodological scalability.

Our key contributions are summarized as follows:

**A systematic framework for activation vectors in ICL.** We formalize an evaluation framework tailored to activation vector methods in ICL paradigm. To the best of our knowledge, we are the first to investigate the application of activation vectors on diverse real-world in-context learning tasks. Within this framework, we adapt and rigorously benchmark two foundational activation vector techniques, revealing critical limitations in their direct applicability to ICL.

**Iterative Vectors: gradient simulation without backpropagation.** We propose Iterative Vectors, a novel activation vector method that uniquely simulates gradient-based optimization without backpropagation. We offer a thoroughly derived theoretical foundation for our method, including both the formulas and the pseudocode, which are closely aligned with the evaluation framework. By taking this approach, we effectively address a significant problem: enhancing language model performance without the need for longer prompts.

**Empirical Validation and Insights.** By iteratively refining task-specific perturbations through aggregation and averaging, IVs achieve an average performance gain of 3.2% over standard ICL baselines. Through further experiments on more challenging tasks, we demonstrate that IVs consistently excel in terms of accuracy, robustness, scalability and speed. Our analysis further uncovers the role of the interactive dynamics in the hyperparameters, demonstrating that IVs conduct principled simulations for gradient-based fine-tuning.

Our code is available on GitHub.

## 2. Related Work

Several studies have investigated the manipulation of language models within the representation space by utilizing lightweight vectors, which we refer to as *activation steering* with *activation vectors* in this paper.

Activation steering methods contrast with existing prompt tuning methods (Li & Liang, 2021; Lester et al., 2021), which operate in a continuous parameter space as part of the prompt and which, crucially, require training via backpropagation.

Again, unlike Parameter-Efficient Fine-Tuning (PEFT) methods, e.g. LoRA (Hu et al., 2021), they do not tune model parameters but instead modify the activations during inference.

### 2.1. Activation Vectors

The term *activation vector* serves as an umbrella concept encompassing lightweight vectors that manipulate language model within their representation space. Depending on implementation specifics, these vectors may interact with FFN layers, attention mechanisms, or both, and their positioning can vary. Task Vectors (Hendel et al., 2023) are extracted from one layer of the model during ICL inference and then applied to a zero-shot query to determine whether they can preserve task-relevant information. Function Vectors (Todd et al., 2023), on the other hand, select activations from the top attention heads based on their causal effect in generating the correct response. These selected activations are then averaged and introduced into a specific layer of the model.

Although these two methods are similar to our approach and objectives, they have primarily been tested on synthetic tasks (e.g., identifying antonyms, naming country capitals, providing plural forms), rather than ICL tasks with demonstrations. Consequently, the practical applicability of these vectors in real-world environments remains uncertain.

In contrast, our objective is to conduct evaluations within a more realistic context by utilizing real-world classification datasets. This approach provides a more thorough framework for assessing activation vectors. We have adapted and included these two methods for comparison to facilitate the practical application of activation vectors beyond theoretical constructs.

### 2.2. Generative Steering

Another research direction focuses on modifying LMs' activations for generation and transfer purposes. Latent Steering Vectors (Subramani et al., 2022) aim at sentence recovery and sentiment transfer. Inference-Time Intervention (Li et al., 2023) involves probing each attention head and guiding the model with the probe vector to enhance the truthfulness of the generated text. Studies by Turner et al. (2023) and Liu et al. (2024) address style and sentiment transfer by employing positive and negative sentence pairs to extract contrastive guidance.

Despite the shared similarities in operating within the representation space, these methods either necessitate training with backpropagation or are restricted to transfer tasks involving sentence pairs. Consequently, it is not immediately clear how they should be integrated into the ICL setting, which we leave for future research.

### 2.3. Cross-modal Vectors

The success of activation vectors has extended their application to visual and multimodal fields. Hojel et al. (2024) analyzed the activations of MAE-VQGAN, identifying acti-

vations that encode task-specific information. Huang et al. (2024) enabled language models to perform multimodal, many-shot in-context learning by utilizing *Multimodal Task Vectors*. Peng et al. (2024) proposed *Learnable In-Context Vector*, which learns task information from demonstrations.

While the referenced research provides valuable interdisciplinary perspectives, this study concentrates on linguistic phenomena. By introducing a novel type of activation vectors, we aim to enhance their analytical potential and create new opportunities for advancement across various fields.

## 3. Method

In this section, we begin by establishing the theoretical foundation of our method. Following this, we outline the evaluation protocols to clearly define the relevant notations. Finally, we present our method in detail.

### 3.1. Theoretical Foundation

Given the significance of in-context learning, numerous theories have been proposed to explain its underlying mechanisms, e.g., Xie et al. (2021); Chan et al. (2022); Ye et al. (2023); Oswald et al. (2023). One particularly intriguing hypothesis posits that a pretrained LM operates as a meta-optimizer, generating meta-gradients which it then applies to address ICL tasks. We now present an overview of this concept.

First, we revisit the dual form of the perceptron and apply it in the modern context of deep neural networks (Irie et al., 2022). Formally, assume a linear layer trained via gradient descent on $T$ training inputs $(\boldsymbol{x}_1, \ldots, \boldsymbol{x}_T)$ and their corresponding (backpropagated) error signals $(\boldsymbol{e}_1, \ldots, \boldsymbol{e}_T)$, where $\boldsymbol{x}_t \in \mathbb{R}^{d_{\text{in}}}$ and $\boldsymbol{e}_t \in \mathbb{R}^{d_{\text{out}}}$. If standard gradient descent is applied, a loss function $\mathcal{L}$ produces the error signal $\boldsymbol{e}_t = -\eta_t(\nabla_{\boldsymbol{y}}\mathcal{L})_t$, where $\eta_t \in \mathbb{R}$ is the learning rate, and $\boldsymbol{y}_t = \boldsymbol{W}\boldsymbol{x}_t$ is the output of the linear layer. Its weight matrix is given by

$$\boldsymbol{W} = \boldsymbol{W}_0 + \sum_{t=1}^{T} \boldsymbol{e}_t \otimes \boldsymbol{x}_t, \qquad (1)$$

where $\boldsymbol{W}_0 \in \mathbb{R}^{d_{\text{out}} \times d_{\text{in}}}$ represents the initial value of the weights. This linear layer transforms an input $\boldsymbol{x} \in \mathbb{R}^{d_{\text{in}}}$ into an output $S_1(\boldsymbol{x}) \in \mathbb{R}^{d_{\text{out}}}$:

$$S_1(\boldsymbol{x}) = \boldsymbol{W}\boldsymbol{x}. \qquad (2)$$

Next, consider a composite layer $S_2$ that stores $T$ key-value pairs, $(\boldsymbol{x}_1, \boldsymbol{e}_1), \ldots, (\boldsymbol{x}_T, \boldsymbol{e}_T)$, represented by a key matrix $\boldsymbol{X} = (\boldsymbol{x}_1, \ldots, \boldsymbol{x}_T) \in \mathbb{R}^{d_{\text{in}} \times T}$ and a value matrix $\boldsymbol{E} = (\boldsymbol{e}_1, \ldots, \boldsymbol{e}_T) \in \mathbb{R}^{d_{\text{out}} \times T}$, along with a weight matrix $\boldsymbol{W}_0 \in \mathbb{R}^{d_{\text{out}} \times d_{\text{in}}}$. This layer transforms an input $\boldsymbol{x} \in \mathbb{R}^{d_{\text{in}}}$ into an

output $S_2(\boldsymbol{x}) \in \mathbb{R}^{d_{\text{out}}}$ by

$$S_2(\boldsymbol{x}) = \boldsymbol{W}_0\boldsymbol{x} + \text{Attn}(\boldsymbol{X}, \boldsymbol{E}, \boldsymbol{x}), \qquad (3)$$

where the parameters of the unnormalized attention operator $\text{Attn}(\cdot)$ are, in order, the key, value, and query.

It can be shown that $S_1$ and $S_2$ are equivalent by expanding the attention operation as

$$\text{Attn}(\boldsymbol{X}, \boldsymbol{E}, \boldsymbol{x}) = \boldsymbol{E}\boldsymbol{X}^\top\boldsymbol{x} = \left(\sum_{t=1}^{T} \boldsymbol{e}_t \otimes \boldsymbol{x}_t\right)\boldsymbol{x}. \qquad (4)$$

This expression elucidates that the forward operation of any linear layer in neural networks, trained via gradient descent, can be interpreted as a key-value-query attention mechanism (Vaswani et al., 2017). In this framework, the training data points act as the keys, the corresponding gradients serve as the values, and the test input generates the query.

Utilizing the dual form, ICL can be interpreted as a meta-optimization process (Dai et al., 2023). This was achieved by reversing the direction of the equivalence and breaking down the attention key and value terms for the ICL query token into its zero-shot and demonstration components, as formally expressed:

$$\widetilde{\mathcal{F}}_{\text{ICL}}(\boldsymbol{q}) = \boldsymbol{W}_{\text{ZSL}}\boldsymbol{q} + \text{LinearAttn}\left(\boldsymbol{W}_V\boldsymbol{X}', \boldsymbol{W}_K\boldsymbol{X}', \boldsymbol{q}\right) \qquad (5)$$

$$= \boldsymbol{W}_{\text{ZSL}}\boldsymbol{q} + \sum_i \boldsymbol{W}_V\boldsymbol{x}_i'\left(\left(\boldsymbol{W}_K\boldsymbol{x}_i'\right)^T\boldsymbol{q}\right) \qquad (6)$$

$$= \boldsymbol{W}_{\text{ZSL}}\boldsymbol{q} + \sum_i \left(\left(\boldsymbol{W}_V\boldsymbol{x}_i'\right) \otimes \left(\boldsymbol{W}_K\boldsymbol{x}_i'\right)\right)\boldsymbol{q} \qquad (7)$$

$$\triangleq \boldsymbol{W}_{\text{ZSL}}\boldsymbol{q} + \Delta\boldsymbol{W}_{\text{ICL}}\boldsymbol{q} \qquad (8)$$

$$= \left(\boldsymbol{W}_{\text{ZSL}} + \Delta\boldsymbol{W}_{\text{ICL}}\right)\boldsymbol{q}. \qquad (9)$$

Here, $\boldsymbol{W}_{\text{ZSL}} = \boldsymbol{W}_V\boldsymbol{X}\left(\boldsymbol{W}_K\boldsymbol{X}\right)^T$ is the zero-shot activation from the static parameters of the model, in which $\boldsymbol{X}$ denotes the input representations of query tokens before the current one, $\boldsymbol{q}$. $\boldsymbol{X}'$ denotes the input representations of the demonstration tokens.

In summary, under the relaxed normalization setting, a pretrained LM acts as a **meta-optimizer**. Through forward computation, the LM generates "meta-gradients" from the demonstration examples, which are then applied to the original parameters via attention, culminating in the formation of the ICL inference capability.

This explanation provides an intuitive understanding of how the LM uses in-context examples. However, it also highlights why ICL performance can be unstable: meta-gradients derived from limited in-context examples may fail to fully capture the task and might not scale appropriately with

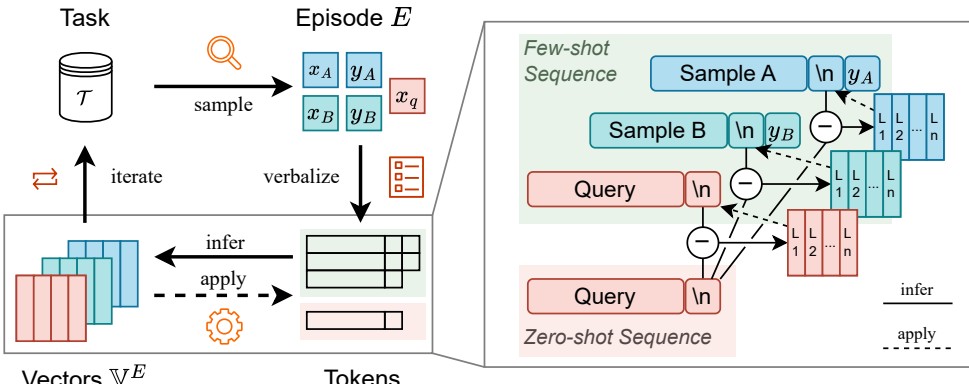

*Figure 2.* An illustration of the extraction and application of Iterative Vectors. By subtracting the zero-shot component, we obtain the meta-gradients that represent the ICL adjustments. These are subsequently refined through successive iterations to obtain the final vectors.

the original parameters. For this reason, we propose Iterative Vectors to extract the meta-gradients—specifically, the activations induced by in-context examples—from the language model's inference process to enhance its accuracy and robustness. This would also allow us to apply these meta-gradients directly in future inference tasks, eliminating the need to compute them afresh from the examples each time a query is evaluated.

Before defining IV, it is necessary to establish the notations employed to evaluate activation vectors.

### 3.2. Activation Vector Evaluation

We adhere to standard few-shot benchmarking protocols (Vinyals et al., 2016; Finn et al., 2017; Snell et al., 2017) to define the activation vector evaluation setting. To assist readers with key terminology and concepts, a comprehensive Glossary is included in Appendix A.

For a given split of an $n$-way $k$-shot classification task $\mathcal{T} = \{\mathcal{T}_{\text{train}}, \mathcal{T}_{\text{val}}, \mathcal{T}_{\text{test}}\}$, which comprises textual query-answer pairs $(x, y)$, an ICL *episode* [1] is sampled as:

$$E = [(x_1, y_1), \ldots, (x_{n \times k}, y_{n \times k}), (x_q, y_q)]. \quad (10)$$

Here, $(x_q, y_q)$ represents the query and its label, preceded by the $n \times k$ support examples. To avoid the impact of unbalanced samples, we uniformly sample $k$ examples from each of the $n$ classes and shuffle them to mitigate any bias arising from sample permutation. We maintain a record of the labels for each example, which can be accessed using $\text{Class}(x_i) \in \{1, 2, \ldots, n, q\}$.

The episode must first be converted into a pure text sequence before the language model $\text{LM}(\cdot)$ can process it. This conversion is handled by a *verbalizer*, which uses a predefined prompt template to instantiate the samples. The template

---

[1] The term is borrowed from meta-learning, considering the meta-gradients at play.

contains two key components: the *input-output separator* that links a question with its answer, and the *example separator* that joins the given support set. To preserve the simplicity of the template, we choose to use one newline (\n) for the input-output separator and three newlines for the example separator, as adopted in Min et al. (2022a).

When the language model $\text{LM}(\cdot)$ is provided with an episode $E$, it performs autoregressive inference on each of the tokens within the verbalized episode. The *clean* prediction of the language model is derived by applying the softmax function to the logits on the potential labels produced by the model, as expressed in the following equation:

$$\hat{y}_{\text{clean}} = \text{LM}(E). \quad (11)$$

In contrast, an *edited* run involves the use of an activation vector editor $f_{\text{edit}}$. The specific method of editing varies based on the chosen approach, and we express the general form as follows:

$$\hat{y}_{\text{edit}} = \text{LM}(E; f_{\text{edit}}(\mathbb{V}, \mathbb{P})), \quad (12)$$

which depends on the set of vectors $\mathbb{V}$ extracted by an *activation vector extractor*, $f_{\text{ext}}$, with hyperparameters $\mathbb{P}$:

$$\mathbb{V} = f_{\text{ext}}(\mathcal{T}_{\text{train}}; \mathbb{P}). \quad (13)$$

The extractor retrieves its target vectors $\mathbb{V}$ from $\mathcal{T}_{\text{train}}$ and identifies the optimal hyperparameters $\mathbb{P}^*$ from $\mathcal{T}_{\text{val}}$ by maximizing the metric $\text{M}$:

$$\mathbb{P}^* = \arg\max_{\mathbb{P}} \text{M}_{E \sim \mathcal{T}_{\text{val}}} (\hat{y}_{\text{edit}}, y_E) \quad (14)$$

$$\mathbb{V}^* = f_{\text{ext}}(\mathcal{T}_{\text{train}}; \mathbb{P}^*). \quad (15)$$

These formulations are presented to illustrate the general principle of activation vector extraction and application, and are therefore intentionally abstract, as their implementation requires flexibility.

For single-token classification tasks, macro-F1, micro-F1, and weighted-F1 scores can serve as the metrics. The vectors $\mathbb{V}^*$ and the optimal hyperparameters $\mathbb{P}^*$ are then applied to the test set $\mathcal{T}_{\text{test}}$ to evaluate the final results $M_{E \sim \mathcal{T}_{\text{test}}}(\hat{y}_{\text{edit}}, y_q)$.

### 3.3. Iterative Vectors

We have demonstrated that attention layers significantly influence ICL, with demonstrations acting as meta-gradients to help the model adapt to the task during inference. We define the extractor $f_{\text{ext}}$ for Iterative Vectors based on this concept.

**Episode Verbalization**  To extract the gradients, we construct two verbalized versions of a given $n$-way $k$-shot episode $E$. The first version, $E = [(x_1, y_1), \ldots, (x_{n \times k}, y_{n \times k}), (x_q, y_q)]$, is the typical shuffled verbalization, which serves as the complete episode. The second version, $E^0 = [(x_q, y_q)]$, is stripped of all demonstrations, resulting in a zero-shot query that provides no information about the task.

Input-output separators are responsible for generating the label words, which gather information and contribute to forming the final prediction (Wang et al., 2023), making the meta-gradients associated with them particularly important. These separators are integral to each $x_i$, specifically as the last token of each, and the subsequent token, denoted as $y_i$, is generated by the language model at the position of the separator token. Given their significance, we use $\text{Act}_l(x_i)$ to denote the activation from the $l$-th attention layer of the $i$-th input-output separator. This generation process causally considers all preceding tokens, including all prior $(x_i, y_i)$ demonstration pairs. Therefore, each $\text{Act}_l(x_i)$ has access to the information contained in all preceding pairs $\{(x_j, y_j) \mid j < i\}$ as well as the current $x_i$.

**Meta-gradient Extraction**  During inference on the two verbalized versions, the extractor collects activations, $\text{Act}_l(x_i)$, for the input-output separator of the $i$-th example in the complete episode $E$, as well as $\text{Act}_l^0(x_q)$ of the query in the zero-shot query $E^0$, across each attention layer $l$ of the LM.

Subsequently, we subtract the zero-shot activations from the complete activations to select and extract the meta-gradients representing the ICL adjustments. Since there are no input-output separators for demonstrations in the zero-shot sequence, all activations from the complete episode use the activations on the input-output separator of the query as the subtrahend:

$$\Delta \text{Act}_l(x_i) = \text{Act}_l(x_i) - \text{Act}_l^0(x_q) \tag{16}$$

When $k > 1$, we average the activations for each class,

resulting in $n$ vectors for each class, plus a vector for the final query:

$$\boldsymbol{v}_l^j = \frac{1}{|\mathbb{C}_j|} \sum_{i \in \mathbb{C}_j} \Delta \text{Act}_l(x_i), \tag{17}$$

$$\boldsymbol{v}_l^q = \Delta \text{Act}_l(x_q) = \text{Act}_l(x_q) - \text{Act}_l^0(x_q), \tag{18}$$

where $\mathbb{C}_j = \{i \mid \text{Class}(x_i) = j\}$. This process yields the meta-gradients for a single episode

$$\mathbb{V}_l^E = \{\boldsymbol{v}_l^1, \boldsymbol{v}_l^2, \ldots, \boldsymbol{v}_l^n, \boldsymbol{v}_l^q\}. \tag{19}$$

By averaging across the task set under extraction, a preliminary version of activation vectors can be obtained.

**Iterative Refinement**  To refine these preliminary vectors, we introduce the concept of **Iterative Vectors (IVs)** by applying the extracted vectors with the editor ($f_{\text{edit}}$) during the extraction phase itself. This emulates standard batched gradient updates. We divide the training set $\mathcal{T}_{\text{train}}$ into $m$ sequential batches of size $b$, denoted $\mathcal{B}_1, \ldots, \mathcal{B}_m$. The iterative extraction process computes a sequence of batch vectors $\mathbb{V}^1, \ldots, \mathbb{V}^m$. These are defined as sets containing the averaged vectors $\{\boldsymbol{v}_l^j, \boldsymbol{v}_l^q\}$ for each layer $l$ and class $j$ or query $q$:

The vectors for the first batch $\mathcal{B}_1$ are calculated by averaging the base vectors $\mathbb{V}_l^E$ from each episode $E \in \mathcal{B}_1$:

$$\mathbb{V}_l^1 = \frac{1}{|\mathcal{B}_1|} \sum_{E \in \mathcal{B}_1} \mathbb{V}_l^E \tag{20}$$

For subsequent batches, the vectors $\mathbb{V}^i$ are computed by averaging episode-specific vectors obtained from episodes $E \in \mathcal{B}_i$, where the language model's activations during extraction are edited by the cumulative average of vectors from all preceding batches, using the extraction strength $\alpha_1$:

$$\mathbb{V}_l^i = \frac{1}{|\mathcal{B}_i|} \sum_{E \in \mathcal{B}_i} \mathbb{V}_l \left[ E | f_{\text{edit}} \left( \bar{\mathbb{V}}_l^{i-1}, \alpha_1 \right) \right], \tag{21}$$

$$\bar{\mathbb{V}}_l^{i-1} = \frac{1}{i-1} \sum_{j=1}^{i-1} \mathbb{V}_l^j, \tag{22}$$

for $i \in \{2, \ldots, m\}$. Here, $\mathbb{V}_l \left[ E | f_{\text{edit}} \left( \bar{\mathbb{V}}_l^{i-1}, \alpha_1 \right) \right]$ denotes the set of vectors extracted from episode $E$ at layer $l$ when activations are modified according to Definition 25 using $\bar{\mathbb{V}}_l^{i-1}$ with strength $\alpha_1$, which will be defined below.

Finally, the extractor $f_{\text{ext}}$ yields the Iterative Vectors by averaging the batch vectors across all iterations:

$$\mathbb{V}_l = \frac{1}{m} \sum_{i=1}^{m} \mathbb{V}_l^i, \tag{23}$$

$$f_{\text{ext}}(\mathcal{T}_{\text{train}}; \mathbb{P}) = \{\mathbb{V}_l \mid l \in \text{LM}\}. \tag{24}$$

| Model | Method | abort. | agnews | athei. | clima. | emoti. | femin. | hate | hilla. | irony | offen. | senti. | sst5 | trec | **Avg.** |
|---|---|---|---|---|---|---|---|---|---|---|---|---|---|---|---|
| | Clean | 32.96 | 53.53 | 25.38 | **27.11** | 24.07 | 31.80 | 49.38 | 35.74 | **55.93** | 51.98 | 36.94 | 29.33 | 64.57 | 39.90 |
| gpt-j-6b | FV | **37.29** | 51.53 | **32.86** | 21.19 | 17.78 | 37.87 | 38.84 | 30.96 | 55.09 | 51.16 | **41.81** | 31.91 | 67.02 | 39.64 |
| | TV | 29.83 | **60.89** | 20.50 | 24.62 | 25.49 | 31.72 | **49.74** | 33.75 | 48.32 | 51.61 | 38.82 | 32.94 | 63.72 | 39.38 |
| | IV (Ours) | 36.06 | 56.13 | 32.05 | 19.23 | **32.70** | **38.20** | 47.30 | **40.68** | 54.65 | 46.32 | 33.17 | **39.07** | **67.32** | **41.76** |
| | Clean | 27.52 | 61.94 | 22.13 | 28.60 | 54.45 | 29.27 | 53.27 | 29.42 | 58.65 | 51.86 | 38.96 | 28.93 | 74.93 | 43.07 |
| llama-2-7b | FV | 25.11 | 67.56 | 14.58 | 23.70 | 58.66 | **31.01** | 52.57 | 32.26 | **60.44** | 54.89 | **42.40** | **30.89** | 71.29 | 43.49 |
| | TV | 27.91 | **72.11** | 21.75 | 31.98 | **59.37** | 29.56 | 50.08 | 29.54 | 50.21 | 52.00 | 41.64 | 29.94 | 74.77 | 43.91 |
| | IV (Ours) | **30.33** | 69.64 | **28.38** | **35.67** | 56.75 | 30.35 | **55.97** | **42.83** | 52.69 | **59.38** | 33.82 | 30.55 | **79.29** | **46.59** |
| | Clean | 29.71 | 79.47 | 13.50 | 19.62 | 69.01 | 34.40 | 53.45 | 40.36 | 52.44 | 56.46 | 38.96 | 36.64 | 74.25 | 46.02 |
| llama-3.1-8b | FV | 29.21 | 83.84 | 15.27 | 18.87 | 68.94 | 34.65 | 55.34 | 34.13 | **55.34** | **56.77** | **47.73** | 36.81 | 72.51 | 46.88 |
| | TV | **30.14** | 80.06 | 13.95 | 15.20 | 68.87 | 28.66 | 53.45 | **43.27** | 52.04 | 56.47 | 39.38 | 36.62 | 74.53 | 45.59 |
| | IV (Ours) | 29.81 | **87.13** | **23.49** | **23.01** | **69.73** | **36.84** | **58.82** | 40.34 | 50.21 | 55.29 | 42.45 | **41.50** | **75.63** | **48.79** |
| | Clean | 34.96 | 76.23 | 27.11 | 20.96 | 61.89 | 37.13 | 53.83 | 45.53 | 55.17 | **60.34** | 38.77 | 38.66 | 76.01 | 48.20 |
| llama-2-13b | FV | **36.55** | 77.37 | 27.25 | 19.71 | 66.73 | 43.35 | 50.57 | **51.16** | 51.26 | 58.94 | **46.15** | 42.72 | 72.57 | 49.56 |
| | TV | 34.71 | 76.28 | 27.24 | 30.88 | 63.27 | 31.87 | 52.63 | 45.03 | 54.98 | 60.14 | 37.82 | 37.98 | 77.05 | 48.45 |
| | IV (Ours) | 35.32 | **79.07** | **27.32** | **38.19** | 67.40 | **46.20** | **57.18** | 50.13 | **66.76** | 59.09 | 35.88 | **44.14** | **80.93** | **52.89** |

*Table 1.* Main experiment results with macro-F1 as the metric. "Clean" denotes a standard one-shot ICL result. The models are GPT-J-6B (Wang & Komatsuzaki, 2021), Llama 2 (Touvron et al., 2023) and Llama 3.1 (Dubey et al., 2024). FV refers to Function Vectors (Todd et al., 2023), while TV denotes Tasks Vectors (Hendel et al., 2023).

**IV Editor** We will now formalize the editor, $f_{\text{edit}}$, for IVs. For the $l$-th attention layer $\text{Attn}_l(\cdot)$, let $\mathbb{V}_l$ denote its corresponding set of extracted IVs. During inference, the editing is performed on each of the input-output separators with the IVs from their corresponding classes, across all layers:

$$\text{EditedAttn}_l(x_i) = \text{Attn}_l(x_i) + \alpha \times \boldsymbol{v}_l^{\text{Class}(x_i)}. \quad (25)$$

This introduces the strength hyperparameter $\alpha$ analogous to learning rates in gradient-based optimization, which controls the update magnitude. Given that meta-gradients tend to be less stable during the iterative process, we have differentiated $\alpha$ into two parameters: the extraction strength $\alpha_1$ and the inference strength $\alpha_2$. These are applied during the iterative extraction and evaluation phases, respectively.

The complete IV hyperparameter set is thus

$$\mathbb{P} = \{k, b, \alpha_1, \alpha_2\}. \quad (26)$$

The extraction shot $k$ controls the number of samples in a sequence during the extraction process. The extraction batch size $b$ serves to replicate a typical batch size used during standard training. The extraction strength $\alpha_1$ denotes the magnitude with which meta-gradients are applied during iterative extraction. Similarly, the inference strength $\alpha_2$ represents the magnitude with which meta-gradients are applied during evaluation. These two parameters share the same notation because they fundamentally represent the same concept, albeit applied in different phases.

The pseudocode for the extraction and evaluation process is available in Appendix B. To facilitate understanding, Appendix C includes an example of the processes described. A comprehensive discussion of hyperparameters is provided in Appendix D.

## 4. Experiments

Iterative Vectors can significantly enhance ICL performance, as demonstrated across four models and 13 diverse tasks (Section 4.1). Furthermore, IVs demonstrate significant time savings in achieving boosted one-shot performance (Section 4.2). They also effectively scale with the quantity of demonstration shots preceding the query (Section 4.3). Whether supplied with only a few or numerous examples for extraction, IVs consistently adapt to the given task, maintaining a trajectory of improved performance (Section 4.4). Finally, through ablating the hyperparameters of our method, we discover an optimal interaction among them that maximizes performance, thereby affirming that each is an essential component of the methodology (Section 4.5).

### 4.1. Main Experiment

We apply our IVs to four popular models across 13 tasks. The results are presented in Table 1. Details of all the datasets used in this paper can be found in Appendix E, while additional results with the other two metrics are provided in Appendix F.

To offer further evidence and a comparative analysis, we adapt two foundational approaches to activation vectors: Function Vectors (Todd et al., 2023) and Task Vectors (Hendel et al., 2023). While these methods were not initially developed for the ICL evaluation setting, we modify them to incorporate the training set by averaging the activations. To ensure a fair comparison, we conduct a search over their respective hyperparameters, as well as the extraction shot parameter $k$. For a detailed overview of their designs, please refer to Appendix G.

During testing, the model cannot ascertain the true class dis-

| Setting | 1-shot | 2-shot | 3-shot | 4-shot | 1-shot + FV | 1-shot + TV | 1-shot + IV (ours) |
|---|---|---|---|---|---|---|---|
| **Macro-F1** | 9.13 | 12.90 | 12.64 | 13.11 | 10.77 | 10.30 | 12.90 |
| **Inference Time** (s) | 1374 | 2434 | 3426 | 4506 | 1389 | 1384 | 1452 |
| **Extraction Time** (min) | - | - | - | - | 438.3 | 14.58 | 23.75 |

*Table 2.* Clean and activation vector results on the *emoji* dataset with model Llama-2-7b. Inference time measurements are based on 10,000 episodes, while extractions are based on 200 episodes.

| Dataset | 2-shot | | | 3-shot | | | 4-shot | | | 5-shot | | |
|---|---|---|---|---|---|---|---|---|---|---|---|---|
| | Clean | +IV | Diff | Clean | +IV | Diff | Clean | +IV | Diff | Clean | +IV | Diff |
| AG News | 76.86 | 79.94 | +3.08 | 80.55 | 82.49 | +1.94 | 82.12 | 84.82 | +2.70 | 82.47 | 85.84 | +3.37 |
| Rotten Tomatoes | 70.28 | 87.50 | +17.22 | 78.97 | 90.57 | +11.60 | 83.74 | 90.74 | +7.00 | 87.80 | 91.48 | +3.68 |

*Table 3.* Multi-shot clean and IV results using the Llama-2-7b model. The displayed metric is macro-F1.

tribution of the test set due to the few-shot setting, which is often imbalanced. Therefore, we adhere to one-shot during the main experiment, [2] which supplies the model with minimal yet sufficient information through a set of uniformly distributed demonstration examples.

We evaluate over 200 episodes for both extraction ($\mathcal{T}_{\text{train}}$) and hyperparameter search ($\mathcal{T}_{\text{val}}$). For the hyperparameters of IVs, we use a fixed iterative batch size of $b = 10$ and explore the extraction strength and inference strength $\alpha_1, \alpha_2 \in \{0.1, 0.3, 0.5, 0.7, 0.9\}$ across all tasks. Regarding the extraction shot $k$, we test $k \in \{1, 2, 3, 4\}$ for both TVs and IVs. However, due to their inherent design, FVs are excessively slow to extract, making it impractical to incorporate additional examples. One significant issue with FV is that it necessitates an extensive search through all attention heads of every layer, posing considerable scaling challenges as the model size grows. Even when limited to $k = 1$, extracting FVs takes approximately 20 times longer than extracting IVs. We present an example of the extraction time required in Table 2.

All experiments were conducted using a predetermined random seed (42) to mitigate selection bias. To ensure a robust representation of result distributions, the tests are averaged over a substantial number of episodes, namely 10,000. All experiments can be performed on a single Nvidia RTX A6000 GPU unless stated otherwise.

The results indicate that Iterative Vectors successfully achieve the goal, surpassing the baselines in most individual tasks as well as in the overall average. For the overall average, Iterative Vectors maintain robust performance gains in all evaluated scenarios, whereas FV/TV exhibit significant performance degradation compared to standard ICL in 37% of the cases (3/8)—a concerning regression that undermines their viability despite their computational overhead. Additionally, IVs show particularly strong results in larger

---

[2]A discussion regarding the decision not to use zero-shot sequences is available in Appendix H.

models. This indicates a promising potential for application to even larger models, as evidenced by our experiments with the Llama-2-70b model, detailed in a supplementary experiment introduced in Section 4.3.

Task Vectors simply identifies the optimal layer for the extraction and application of vectors, which can serve as a simple baseline for future research. Although Function Vectors achieve relatively better results than Task Vectors, their high search time presents significant challenges for optimization in practical ICL applications.

### 4.2. IVs Save Inference Time

All the aforementioned experiments require only a single demonstration example during application, demonstrating that activation vectors can significantly reduce inference time. To highlight this point, we turn to the *emoji* dataset, a 20-class classification task. Evaluating this dataset with multi-shot demonstrations would be exceedingly time-consuming due to the rapid increase in the length of the demonstration sequence.

We apply IV on this dataset and further fix the extraction shot at $k = 1$ rather than exploring the range $k = \{1, 2, 3, 4\}$ to further minimize the time required for hyperparameter search.

The results, presented in Table 2, clearly show that IVs substantially enhance performance with minimal time expenditure, in stark contrast to higher-shot ICL cases, which required significantly more time. In 2-shot and 3-shot settings, the inference times are 2,434s and 3,426s, respectively. However, IV achieves the same score of 12.90 as the 2-shot setting in 1,452s—41% faster—and exceeds the 3-shot performance (12.64). Crucially, IV reduces feature extraction time by 95% compared to FV (23.75 min vs 438.3 min), while maintaining a 20% performance advantage over TV (12.90 vs 10.30). While TV's naive architecture enables rapid extraction, this approach exhibits catastrophic failure in 50% of the model averages, as previously shown. This

positions IV as the most time-effective solution, balancing accuracy with practical computational demands.

## 4.3. IVs Scale with In-Context Demonstrations

One might wonder why activation vectors are not applied to higher-shot settings. The primary reason is that a key objective of using activation vectors is to reduce the prompt length and the inference time associated with higher-shot scenarios. Nonetheless, we conducted experiments to evaluate their performance with longer demonstrations.

For this study, we have chosen the *AG News* and *Rotten Tomatoes* datasets. This selection is based on the observation that the language model under evaluation demonstrates progressively improved performance as the number of examples increases, as illustrated in Table 3. Consequently, this poses a more substantial challenge for the IVs to improve upon. However, the results demonstrate that IVs scale effectively with the number of demonstration shots preceding the query. IV boosts high baselines while maintaining ICL compatibility, delivering +3.37% on AG News' near-peak 5-shot (82.47% to 85.84%) and +17.22% on Rotten Tomatoes' 2-shot (70.28% to 87.50%). Consistent gains across all shots (1-5) confirm its compatibility with the ICL framework. This suggests that IVs can offer advantages even when initial performance levels are already high, and they integrate seamlessly with the ICL framework.

In addition, one could contemplate a similar challenge using larger models. The results are comparable; please refer to Table 8, where the improvement of IVs is once again evident with Llama-2-70b.

## 4.4. IVs Improve with Increased Extraction Episodes

An important aspect to consider is the number of examples required for IVs to function effectively. We design an experiment to test various numbers of extraction episodes, which in turn controls the number of examples used to extract the IVs. Additionally, another critical aspect is the stability of IVs when extracted from different numbers of episodes. To evaluate this, we reuse hyperparameters obtained from prior searches in the main experiment ($k = 4$, $b = 10$ fixed, $\alpha_1 = 0.3$, $\alpha_2 = 0.5$), rather than optimizing hyperparameters for each different episode count. The results are presented in Table 4.

The data shows that, IVs surpass ICL prompt limitations through scalable example utilization, delivering consistent gains with $\geq 3$ episodes (+0.57–7.6%) even under fixed, potentially suboptimal hyperparameters. While minor fluctuations occur with a smaller number of episodes ($\leq 2$), performance consistently improves with an increased volume of examples. This demonstrates the IVs' capability to extract and utilize a larger number of examples, thereby

| Episodes | Clean | 1 | 2 | 3 | 5 | 10 |
|---|---|---|---|---|---|---|
| Macro-F1 | 62.15 | 40.64 | 54.44 | 62.72 | 66.17 | 64.27 |

| Episodes | 20 | 30 | 50 | 100 | 150 | 200 |
|---|---|---|---|---|---|---|
| Macro-F1 | 63.01 | 65.05 | 66.77 | 68.14 | 69.71 | 69.62 |

*Table 4.* IV results with different number of extraction episodes, using a fixed set of hyperparameters. The model utilized is Llama-2-7b, and the dataset is AG News.

surpassing the traditional limits of ICL.

## 4.5. Ablation Study

We conduct an ablation study to analyze the impact of key hyperparameters on our method's performance and dynamics. We focus particularly on the extraction batch size $b$, which was set to $b = 10$ in the experiments presented earlier. We analyze the results on the validation set, which is used to determine the optimal hyperparameters during tuning. The results are visualized in Figure 3.

Varying the extraction batch size $b$ reveals insights into the iterative refinement process.

**No Iteration** ($b = 0$)   When $b = 0$, the iterative update mechanism is effectively disabled. Extracted vectors are not reintroduced into the model to influence subsequent extractions. This configuration results in significantly poorer performance compared to any setting where $b \geq 1$. Furthermore, without editing during extraction, the extraction strength $\alpha_1$ becomes irrelevant as there is no process for it to control.

**Single-Episode Iteration** ($b = 1$)   Setting $b = 1$ means each "batch" consists of a single episode. While not true batching, this setting does enable the iterative refinement process where vectors extracted from one episode are used to edit the extraction from the next episode. This results in a substantial performance boost over the $b = 0$ case. This underscores the critical importance and contribution of the core *Iterative* Vectors concept – the process of reintroducing and refining vectors across episodes, even without explicit multi-episode batch averaging before application.

**Increasing Batch Size** ($b > 1$)   As the extraction batch size $b$ increases beyond 1, we observe a general trend of improved performance. Examining the optimal hyperparameter pairs ($\alpha_1$, $\alpha_2$) found during the validation search, we see that for small batch sizes ($b \geq 1$), optimal solutions often involve a high extraction strength ($\alpha_1$) and a relatively low inference strength ($\alpha_2$). This suggests that vectors extracted from smaller batches are less stable and require strong iterative refinement ($\alpha_1$) but must be applied cautiously during inference ($\alpha_2$). As $b$ increases, allowing averaging over larger groups of episodes before each itera-

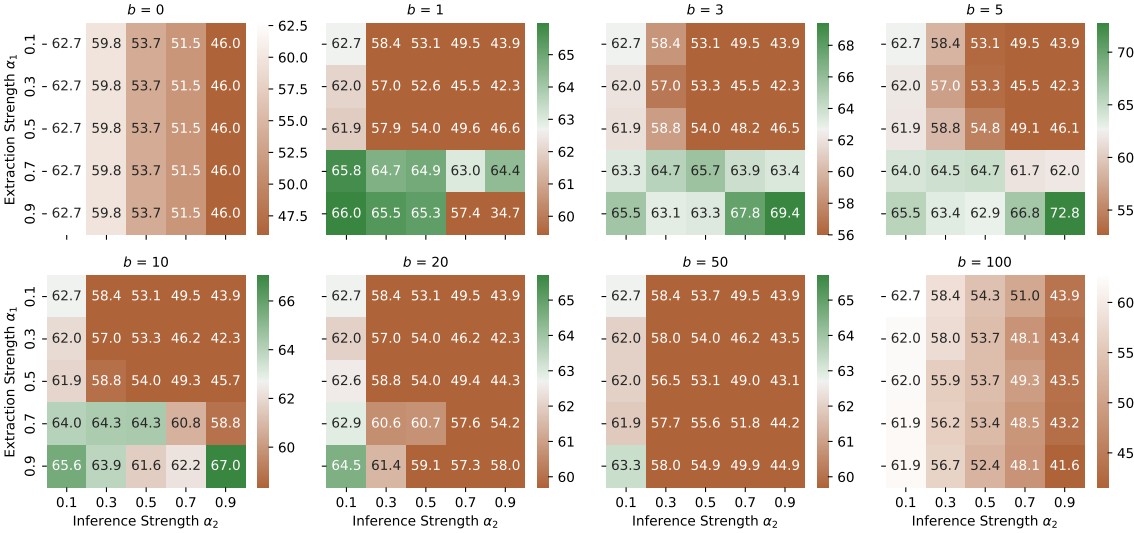

*Figure 3.* Ablation study on the hyperparameters. The model utilized is Llama-2-7b, and the dataset evaluated is the validation split of AG News, with macro-F1 serving as the metric. Note that $b = 0$ indicates the absence of iterative refining and batching.

tive update step, the extracted vectors become more stable. This increased stability is reflected in the optimal $\alpha_2$ values, which tend to rise, enabling the method to leverage stronger inference strengths for better performance.

**Excessively Large Batch Size**  While increasing batch size generally helps stabilize vectors, Figure 3 also indicates diminishing returns. If the batch size becomes excessively large, performance may plateau or even decrease. We hypothesize that very large batches lead to fewer total iterative updates over the training data (as $m$ decreases), thereby reducing the opportunities for the vectors to be iteratively refined across different subsets of the training distribution. This suggests a trade-off between the stability gained by averaging over a larger batch and the benefits of more frequent iterative refinement steps.

These observed interactions between the extraction batch size ($b$), the extraction strength ($\alpha_1$), and the inference strength ($\alpha_2$) clearly demonstrate how each hyperparameter plays a crucial role in the overall dynamics and effectiveness of the Iterative Vectors methodology. For a more comprehensive discussion, including guidance on tuning these and other hyperparameters, please refer to Appendix D.

## 5. Conclusion

In this work, we introduced Iterative Vectors (IVs), a novel method for enhancing In-Context Learning (ICL) by refining task-specific activation differences directly in a language model's activation space—without backpropagation or parameter updates. Our key innovation is an iterative refinement process: by splitting the extraction into batches and

successively refining each batch's result with previous ones, IVs achieve greater stability and effectiveness than prior activation steering methods like Function Vectors (FV) and Task Vectors (TV).

Comprehensive experiments across four language models and 13 real-world tasks consistently showed IVs outperform standard ICL and existing baselines, with ablation studies confirming the iterative process's importance and the influence of batch size and application strengths. These findings highlight IVs as a resource-efficient, practical alternative to fine-tuning or prompt engineering, opening promising avenues for improving and adapting large language models by leveraging their rich activation spaces.

## Impact Statement

This paper presents work whose goal is to advance the field of Machine Learning. There are many potential societal consequences of our work, none which we feel must be specifically highlighted here.

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

# A. Glossary

**Act**$(x_i)$ Notation used to represent the activation vector extracted at the position of the input-output separator token for the $i$-th example ($x_i$) within a complete verbalized episode, capturing information from all preceding tokens.

**Act**$^0(x_q)$ Notation used to represent the activation vector extracted at the position of the input-output separator token for the query ($x_q$) within a zero-shot verbalized version of the episode (containing only the query).

**Activation Vector(s)** ($\mathbb{V}$) A general term encompassing lightweight vectors derived from language model activations that are used to manipulate the model's behavior, typically by adding or modifying activations during inference. Iterative Vectors (IVs) are a specific type of activation vector.

**Activation Vector Editor** ($f_{\textbf{edit}}$) A function or mechanism that applies extracted activation vectors to the language model's internal state (e.g., attention activations) during inference to steer its behavior.

**Activation Vector Extractor** ($f_{\textbf{ext}}$) A function or process that derives activation vectors ($\mathbb{V}$) from a given dataset or set of examples, often involving hyperparameters ($\mathbb{P}$). The final Iterative Vectors are the output of the extractor $f_{\text{ext}}(\mathcal{T}_{\text{train}}; \mathbb{P})$.

**Batch** ($\mathcal{B}_i$) In the context of Iterative Vector extraction, a subset of the training episodes used in a single step of the sequential iterative process. The full training set is divided into $n$ sequential batches $\mathcal{B}_1, \ldots, \mathcal{B}_n$.

**Batch Vectors** ($\mathbb{V}^i$) The set of activation vectors obtained by averaging the episode-specific vectors ($\mathbb{V}_l^E$) from all episodes within a single batch $\mathcal{B}_i$. In the iterative process, these batch vectors are computed sequentially, with extraction for batch $i$ potentially being edited by the cumulative average of vectors from previous batches.

**Clean Prediction** ($\hat{y}_{\textbf{clean}}$) The standard output or prediction of the language model when processing an episode or query without any modification by an activation vector editor.

$\Delta \text{Act}_l(x_i)$ The difference between the activation at layer $l$ for the input-output separator of the $i$-th example in the complete episode and the activation at layer $l$ for the input-output separator of the query in the zero-shot episode. Represents the ICL-induced activation difference.

**Edited Run** ($\hat{y}_{\textbf{edit}}$) An inference pass where an activation vector editor modifies the language model's internal activations based on extracted activation vectors.

**Episode** ($E$) In the context of activation vector evaluation, a single sample for ICL consisting of a set of support examples $(x_i, y_i)$ followed by a query example $x_q$, arranged in a specific sequence.

**Example Separator** A token or sequence of tokens used in a verbalizer's template to separate individual demonstration examples from each other and from the query example within a prompt.

**Extraction Batch Size** ($b$) A hyperparameter in the iterative refinement process specifying the number of episodes included in each batch ($\mathcal{B}_i$) used during iterative extraction.

**Extraction Episode** ($k$) A hyperparameter specifying the number of support examples used in each episode sampled from the training set during the activation vector extraction process.

**Extraction Strength** ($\alpha_1$) A hyperparameter controlling the magnitude with which the cumulative average of previously extracted vectors ($\bar{\mathbb{V}}^{i-1}$) is applied to edit activations during the iterative extraction process.

**Inference Strength** ($\alpha_2$) A hyperparameter controlling the magnitude with which the final, extracted Iterative Vectors ($f_{\text{ext}}(\mathcal{T}_{\text{train}}; \mathbb{P})$) are applied to edit activations during the evaluation phase on the validation or test set.

**Input-Output Separator** A token or sequence of tokens used in a verbalizer's template to link an input (e.g., a question) with its corresponding output (e.g., an answer) within a demonstration or query example.

$\mathbb{V}_l^E$ The set of class-specific ($\boldsymbol{v}_l^j$) and query ($\boldsymbol{v}_l^q$) activation vectors derived from a single episode $E$ at layer $l$ using the base subtraction method.

$\bar{\mathbb{V}}^{i-1}$ In the iterative extraction process for batch $i$ ($i \geq 2$), this refers to the average of the vectors $\mathbb{V}^1, \ldots, \mathbb{V}^{i-1}$ computed from all preceding batches. This cumulative average is used to edit activations when extracting vectors from episodes in batch $i$.

$\mathbb{V}_l\left[E|f_{\textbf{edit}}\left(\bar{\mathbb{V}}_l^{i-1},\alpha_1\right)\right]$  The set of vectors extracted from a single episode $E$ at layer $l$, obtained when the language model's activations during the extraction forward pass are modified (edited) using the cumulative average of vectors from preceding batches ($\bar{\mathbb{V}}_l^{i-1}$) with the extraction strength ($\alpha_1$).

$\boldsymbol{v}_l^j$  The average $\Delta\,\text{Act}_l$ vector for all examples belonging to class $j$ within an episode, representing the aggregated meta-gradient for that class at layer $l$.

$\boldsymbol{v}_l^q$  The $\Delta\,\text{Act}_l$ vector for the query example, representing the meta-gradient associated with the query itself at layer $l$.

**Verbalizer**  A component that converts structured data (like an ICL episode) into a pure text sequence suitable for a language model, typically using a predefined prompt template with input-output and example separators.

## B. Pseudocode

We present the extraction process in Algorithm 1. The evaluation procedures are defined in Algorithm 2. The episodic functions employed in the processes can be found in Algorithm 3.

For information on our hyperparameters, please refer to the extraction shot $k$, batch size $b$, and strength $\alpha_1$ as detailed in Algorithm 1. The inference strength, denoted as $\alpha_2$, is used in Algorithm 2.

## C. Example

For instance, consider a 2-way (Positive/Negative) 2-shot task. A verbalized episode ($E$) might look like this (using newline for input-output separator and triple newline for example separator):

```
great movie \n Positive \n\n\n
terrible book \n Negative \n\n\n
hated it \n Negative \n\n\n
amazing film \n Positive \n\n\n
loved that movie \n
```

The zero-shot query ($E^0$) for the query input "loved that movie" is:

```
loved that movie \n
```

Here, there are four support examples (two Positive, two Negative), followed by the query "loved that movie $\backslash$n". During inference on the complete episode $E$, we collect activations at the input-output separator token (the newline after "Positive", "Negative", "Negative", "Positive", and the newline after "loved that movie"). This is done for each attention layer $l$. In parallel, we collect the activation at the input-output separator token (the newline after "loved that movie") from the zero-shot version $E^0$. Let these activations be:

- $\text{Act}_l$("great movie"): Activation for the separator after the 1st support input (Positive) in $E$.

- $\text{Act}_l$("terrible book"): Activation for the separator after the 2nd support input (Negative) in $E$.

- $\text{Act}_l$("hated it"): Activation for the separator after the 3rd support input (Negative) in $E$.

- $\text{Act}_l$("amazing film"): Activation for the separator after the 4th support input (Positive) in $E$.

- $\text{Act}_l$("loved that movie"): Activation for the separator after the query input in $E$.

- $\text{Act}_l^0$("loved that movie"): Activation for the separator after the query input in $E^0$.

Subsequently, we subtract the zero-shot activation $\text{Act}_l^0$("loved that movie") from the complete activations to isolate the

ICL-induced component. For our 2-shot example, this yields five $\Delta \operatorname{Act}_l$ vectors at layer $l$:

$$\Delta \operatorname{Act}_l(\text{``great movie''}) = \operatorname{Act}_l(\text{``great movie''}) - \operatorname{Act}_l^0(\text{``loved that movie''})$$
$$\Delta \operatorname{Act}_l(\text{``amazing film''}) = \operatorname{Act}_l(\text{``amazing film''}) - \operatorname{Act}_l^0(\text{``loved that movie''})$$
$$\Delta \operatorname{Act}_l(\text{``terrible book''}) = \operatorname{Act}_l(\text{``terrible book''}) - \operatorname{Act}_l^0(\text{``loved that movie''})$$
$$\Delta \operatorname{Act}_l(\text{``hated it''}) = \operatorname{Act}_l(\text{``hated it''}) - \operatorname{Act}_l^0(\text{``loved that movie''})$$
$$\Delta \operatorname{Act}_l(\text{``loved that movie''}) = \operatorname{Act}_l(\text{``loved that movie''}) - \operatorname{Act}_l^0(\text{``loved that movie''})$$

These $\Delta \operatorname{Act}_l$ values represent the change in activation space attributable to the presence of the demonstration example(s) compared to the zero-shot baseline, specific to each example in the episode.

As $k > 1$, Equation 17 involves averaging the $\Delta \operatorname{Act}_l$ vectors for all examples within a specific class $j$ to obtain the class-specific meta-gradient vector $\boldsymbol{v}_l^j$. For our 2-way 2-shot example, we have two positive and two negative examples:

$$\boldsymbol{v}_l^{\text{Positive}} = \frac{1}{2} \left( \Delta \operatorname{Act}_l(\text{``great movie''}) + \Delta \operatorname{Act}_l(\text{``amazing film''}) \right)$$
$$\boldsymbol{v}_l^{\text{Negative}} = \frac{1}{2} \left( \Delta \operatorname{Act}_l(\text{``terrible book''}) + \Delta \operatorname{Act}_l(\text{``hated it''}) \right)$$

The query vector $\boldsymbol{v}_l^q$ is simply the $\Delta \operatorname{Act}_l$ calculated for the query example:

$$\boldsymbol{v}_l^q = \Delta \operatorname{Act}_l(\text{``loved that movie''})$$

This averaging aggregates the meta-gradients from multiple instances of the same class, yielding the set of vectors for this episode: $\mathbb{V}_l^E = \{\boldsymbol{v}_l^{\text{Positive}}, \boldsymbol{v}_l^{\text{Negative}}, \boldsymbol{v}_l^q\}$.

The iterative refinement process emulates batched gradient updates. Consider a total of 10 extraction episodes and a batch size $b = 2$, which gives us $m = 5$ batches.

1. We process the first batch of $b = 2$ episodes ($\mathcal{B}_1$). Extract $\mathbb{V}_l^1$ from these episodes *without* editing. $\mathbb{V}_l^1$ is the average of the $\mathbb{V}_l^E$ sets derived from the two episodes in $\mathcal{B}_1$.

2. We then process the second batch of $b = 2$ episodes ($\mathcal{B}_2$). During the extraction pass for each episode in $\mathcal{B}_2$, we use the already computed vectors $\mathbb{V}_l^1$ with strength $\alpha_1$ to edit the model's activations – specifically, by adding the relevant class/query vector from $\mathbb{V}_l^1$ to the corresponding separator activations. The extractor then computes $\mathbb{V}_l^E$ for each episode in $\mathcal{B}_2$ based on these *edited* runs, and averages them to produce $\mathbb{V}_l^2$.

3. This continues for batches $i = 3, \ldots, m$. During the extraction pass for each episode in batch $i$, the model's activations are edited using the cumulative average of the vectors obtained from all preceding batches ($\bar{\mathbb{V}}_l^{i-1} = \frac{1}{i-1} \sum_{j=1}^{i-1} \mathbb{V}_l^j$) with strength $\alpha_1$. The extractor computes $\mathbb{V}_l^E$ for each episode in batch $i$ based on these edited runs, and averages them to produce $\mathbb{V}_l^i$.

4. Finally, the extractor $f_{\text{ext}}$ averages the vectors obtained from all batches: $\frac{1}{5}(\mathbb{V}_l^1 + \mathbb{V}_l^2 + \mathbb{V}_l^3 + \mathbb{V}_l^4 + \mathbb{V}_l^5)$. This yields the refined set of IVs for each layer.

This process allows the vectors to be refined based on previous iterations, analogous to how model weights are updated in batches during training.

## D. Hyperparameters of IV

In this paper, we introduce four hyperparameters: the extraction shot $k$, the extraction batch size $b$, the extraction strength $\alpha_1$, and the inference strength $\alpha_2$. These notations have been used consistently throughout the paper, including in formulas, pseudocode, and explanations. We now provide a detailed discussion of each hyperparameter and its function, followed by a guide on how to tune them effectively.

**The extraction shot $k$ controls the number of samples in a sequence during the extraction process.** This originates from the definition of an $n$-way $k$-shot episode (Eq. 10). During extraction, a longer support sequence may enhance the model's understanding of the task, thereby producing higher-quality meta-gradients. However, since adding more samples does not guarantee improved performance, and a larger $k$ increases extraction time, we propose optimizing this hyperparameter through a search process.

**The extraction batch size $b$ serves to replicate a typical batch size used during standard training.** As implemented in Algorithm 1, the preliminary vectors extracted are averaged every $b$ episodes to form the Iterative Vectors, which are subsequently incorporated into the extraction process. Unlike previous approaches that defer meta-gradient application, our method introduces them during extraction, enabling immediate feedback to the model's hidden states. This iterative refinement mechanism ensures that meta-gradients actively guide each forward pass of the LM, progressively refining intermediate representations and amplifying their influence on subsequent episodes. The result is a self-reinforcing cycle where meta-gradients sharpen the contrast between zero-shot sequences and iteratively improve their own quality, driving robust representation learning.

In Section 4.5, we empirically validate the critical role of batch size $b$ through ablation studies, demonstrating that optimal $b$ selection yields measurable performance gains.

**The extraction strength $\alpha_1$ denotes the magnitude with which meta-gradients are applied during iterative extraction.** Similarly, the inference strength $\alpha_2$ represents the magnitude with which meta-gradients are applied during evaluation. These two parameters share the same notation because they fundamentally represent the same concept, albeit applied in different phases.

In the application of vectors, all methods evaluated in this paper utilize vector addition. However, the meta-gradients may not scale properly with the original parameters. Therefore, we propose scaling them before incorporating them into the hidden states, a consideration not derived from nor addressed in previous methods. During the iterative extraction phase, the scaling constant is $\alpha_1$, whereas during evaluation, the constant is $\alpha_2$.

We differentiate the strength into two parameters because meta-gradients are less stable during the iterative process. This instability can accumulate across layers and episodes, so we aim to apply a lower strength during extraction, if necessary, to mitigate this issue.

**Guide to tuning the hyperparameters.** We recommend a higher value of $k$ for tasks in which the LM demonstrates greater proficiency. Exploring the range of $k \in \{1, 2, 3, 4\}$ is both straightforward and effective, as demonstrated in our experiments.

Concerning batch size, we have demonstrated that it should neither be too large nor too small. We recommend starting with $b = 5$ or $b = 10$. Methods for tuning typical batch sizes may also be considered.

Regarding the strength parameters $\alpha_1$ and $\alpha_2$, we performed a comprehensive grid search within the range $[0.1, 0.9]$. Future research is encouraged to employ more sophisticated search strategies, as these parameters often cluster in a low-performance consecutive area (see Figure 3), which can be pruned if properly identified.

# E. Datasets

A full list of all datasets utilized in this research, along with their corresponding access labels, is detailed in Table 5. The datasets are obtained from HuggingFace (Lhoest et al., 2021).

AG News (Zhang et al., 2015) is a subdataset of AG's corpus of news articles constructed by assembling titles and description fields of articles from the 4 largest classes ("World", "Sports", "Business", "Sci/Tech") of AG's Corpus.

TweetEval (Barbieri et al., 2020) introduces an evaluation framework consisting of a series of Twitter-specific classification tasks. We selected all single-token classification tasks from the dataset.

The Rotten Tomatoes dataset (Pang & Lee, 2005) is a collection of movie reviews and ratings from the Rotten Tomatoes website, often used for sentiment analysis and natural language processing tasks.

The SST5 dataset, derived from the Stanford Sentiment Treebank (Socher et al., 2013), is a collection of movie reviews

| Name | Abbr. Used | Huggingface Label |
|------|-----------|-------------------|
| Abortion | abor. | tweet_eval/stance_abortion |
| AG News | agnews | ag_news |
| Atheism | athe. | tweet_eval/stance_atheism |
| Climate | clim. | tweet_eval/stance_climate |
| Emoji | - | tweet_eval/emoji |
| Emotion | emot. | tweet_eval/emotion |
| Feminist | femi. | tweet_eval/stance_feminist |
| Hate | hate | tweet_eval/hate |
| Hillary | hill. | tweet_eval/stance_hillary |
| Irony | irony | tweet_eval/irony |
| Offensive | offe. | tweet_eval/offensive |
| Rotten Tomatoes | - | rotten_tomatoes |
| Sentiment | sent. | tweet_eval/sentiment |
| SST 5 | sst5 | SetFit/sst5 |
| TREC | trec | trec |

*Table 5.* The datasets and tasks employed, along with their corresponding abbreviations used in the result tables, and their respective labels as hosted on Hugging Face.

| Model | Task | abort. | agnews | athei. | clima. | emoti. | femin. | hate | hilla. | irony | offen. | senti. | sst5 | trec | Avg. |
|-------|------|--------|--------|--------|--------|--------|--------|------|--------|-------|--------|--------|------|------|------|
| gpt-j-6b | Clean | 39.17 | 57.97 | 30.49 | 30.92 | 31.91 | 37.70 | 49.39 | 40.33 | **59.86** | **63.22** | 38.73 | 32.62 | 68.23 | 44.66 |
| | FV | 51.93 | 55.39 | **45.81** | 24.89 | 29.62 | **54.20** | 45.48 | **58.97** | 57.30 | 58.25 | **41.77** | 37.37 | **69.70** | **48.51** |
| | TV | 51.52 | **65.86** | 23.72 | **32.84** | 32.85 | 37.64 | **49.74** | 37.89 | 48.32 | 60.05 | 40.23 | 35.60 | 64.75 | 44.69 |
| | IV (Ours) | **60.02** | 61.30 | 44.59 | 20.49 | 37.36 | 49.05 | 48.32 | 55.29 | 56.30 | 46.94 | 34.48 | **40.08** | 67.32 | 47.81 |
| llama-2-7b | Clean | 28.69 | 63.40 | 24.90 | 34.88 | 57.31 | 30.25 | 53.64 | 30.05 | 62.22 | 53.67 | 40.02 | 43.08 | **77.33** | 46.11 |
| | FV | 30.25 | 69.56 | 18.50 | 25.49 | **62.91** | 36.07 | 57.16 | 35.29 | 63.83 | 63.95 | **46.44** | 45.22 | 75.54 | 48.48 |
| | TV | 29.31 | **72.97** | 24.50 | **62.14** | 62.52 | 30.47 | 50.09 | 30.14 | 52.86 | 53.53 | 41.07 | 43.28 | 77.10 | 48.46 |
| | IV (Ours) | **35.88** | 72.45 | **39.17** | 58.46 | 58.96 | **40.03** | 58.46 | **48.83** | 53.01 | 63.59 | 36.25 | **46.67** | 76.83 | **52.97** |
| llama-3.1-8b | Clean | 39.18 | 80.64 | 18.14 | 21.26 | 74.06 | 47.17 | 53.66 | 48.14 | 53.96 | 60.12 | 39.01 | 45.25 | 69.69 | 50.02 |
| | FV | 41.93 | 84.31 | 21.15 | 20.47 | 74.35 | 51.76 | 55.45 | 44.08 | **56.06** | **69.89** | 48.32 | 42.43 | 68.20 | 52.18 |
| | TV | 39.07 | 81.12 | 18.55 | 20.21 | 74.47 | 40.21 | 53.47 | 50.33 | 53.67 | 60.35 | 39.13 | 43.04 | 69.62 | 49.48 |
| | IV (Ours) | **44.25** | **87.30** | **36.33** | **22.33** | **77.70** | **56.57** | **58.84** | **56.07** | 52.23 | 69.20 | 42.83 | **48.85** | **70.24** | **55.60** |
| llama-2-13b | Clean | 52.57 | 77.96 | 42.78 | 20.36 | 65.42 | 55.94 | 54.00 | 56.83 | 55.19 | 63.56 | 41.41 | 44.44 | **78.56** | 54.54 |
| | FV | 53.16 | 78.81 | **48.92** | 19.57 | 69.99 | **64.96** | **58.94** | 62.25 | 52.32 | 70.70 | **47.87** | **49.19** | 76.58 | 57.94 |
| | TV | 51.34 | 78.07 | 43.22 | 49.38 | 67.27 | 47.60 | 53.22 | 56.05 | 55.05 | 62.82 | 39.70 | 43.86 | 76.16 | 55.67 |
| | IV (Ours) | **55.67** | **80.33** | 46.74 | **65.56** | **71.03** | 58.84 | 58.67 | **63.13** | **66.96** | **73.80** | 36.74 | 47.90 | 77.47 | **61.76** |

*Table 6.* Main experiment results with micro-F1 as the metric. "Clean" denotes a standard one-shot ICL result.

annotated with fine-grained sentiment labels, offering a five-class sentiment classification task ranging from very negative to very positive.

Text Retrieval Conference Question Answering (TrecQA) (Wang et al., 2007) is a dataset created from the TREC-8 (1999) to TREC-13 (2004) Question Answering tracks.

Our few-shot evaluation methodology employs episodic sampling to regulate the duration of both extraction and inference processes, rather than relying solely on the absolute number of samples. Consequently, not all available samples are utilized during the experimental procedures. This aspect underscores an additional dimension of efficiency inherent in activation vectors.

# F. Additional results

We present the results of our main experiment on the other two metrics, namely micro-F1 and weighted-F1, derived from our main experiment, in Table 6 and Table 7, respectively.

According to these evaluation criteria, IV outperforms both FV and TV in the majority of tasks, consistently achieving a higher average score. The only exception occurs in the GPT-J-6B and micro-F1 setting (Table 6), where FV demonstrates

| Model | Task | abort. | agnews | athei. | clima. | emoti. | femin. | hate | hilla. | irony | offen. | senti. | sst5 | trec | **Avg.** |
|---|---|---|---|---|---|---|---|---|---|---|---|---|---|---|---|
| gpt-j-6b | Clean | 42.61 | 53.69 | 34.82 | **34.83** | 22.48 | 40.34 | 49.46 | 42.14 | **58.64** | 62.47 | 33.50 | 31.82 | 68.12 | 44.22 |
| | FV | 52.83 | 51.62 | **50.11** | 31.38 | 17.29 | **52.93** | 35.96 | 47.47 | 57.23 | 59.41 | **39.82** | 35.19 | **69.86** | 46.24 |
| | TV | 49.48 | **61.07** | 26.01 | 34.19 | 22.74 | 40.30 | **49.79** | 39.49 | 48.21 | 60.68 | 34.78 | 35.52 | 65.18 | 43.65 |
| | IV (Ours) | **56.37** | 56.16 | 48.98 | 15.48 | **33.59** | 50.39 | 46.26 | **52.34** | 56.49 | 48.88 | 32.98 | **40.08** | 68.38 | **46.64** |
| llama-2-7b | Clean | 30.58 | 62.03 | 27.50 | 38.72 | 57.45 | 31.75 | 53.83 | 27.79 | 61.15 | 56.07 | 35.33 | 34.46 | 77.58 | 45.71 |
| | FV | 31.40 | 67.69 | 16.00 | 25.62 | **62.86** | 38.41 | 54.68 | 33.09 | **62.93** | 63.85 | 35.83 | 36.79 | 77.29 | 46.65 |
| | TV | 31.43 | **72.23** | 27.39 | **60.09** | 62.70 | 32.06 | 50.00 | 27.66 | 52.57 | 55.85 | **39.36** | 35.39 | 77.27 | 48.00 |
| | IV (Ours) | **38.90** | 69.75 | **44.22** | 59.10 | 59.02 | **41.32** | **57.46** | **50.01** | 51.86 | **65.18** | 27.70 | 36.94 | **78.22** | **52.28** |
| llama-3.1-8b | Clean | 40.92 | 79.57 | 15.32 | **13.97** | 73.77 | 47.66 | 53.04 | 48.62 | 50.70 | 62.16 | 36.04 | 40.44 | 70.66 | 48.68 |
| | FV | 43.03 | 83.91 | 20.32 | 10.22 | 74.01 | 50.30 | 55.02 | 43.71 | **54.11** | 67.33 | **44.67** | 38.50 | 70.74 | 50.45 |
| | TV | 41.06 | 80.17 | 16.45 | 9.35 | 73.86 | 41.34 | 53.33 | 51.20 | 50.23 | 62.30 | 36.09 | 39.41 | 70.65 | 48.11 |
| | IV (Ours) | **44.98** | **87.18** | **39.73** | 11.41 | **76.67** | **53.66** | **58.70** | **54.28** | 48.05 | 66.34 | 38.88 | **44.27** | **72.86** | **53.62** |
| llama-2-13b | Clean | 51.80 | 76.36 | 45.57 | 19.77 | 65.73 | 53.00 | 53.46 | 55.25 | 54.99 | 65.44 | 33.47 | 41.63 | 79.10 | 53.51 |
| | FV | 52.92 | 77.47 | **49.87** | 22.99 | 70.76 | **60.23** | 53.47 | **60.28** | 49.71 | 68.68 | **41.76** | 46.51 | 78.98 | 56.43 |
| | TV | 51.32 | 76.43 | 45.95 | 51.92 | 67.44 | 46.91 | 51.91 | 54.67 | 54.63 | 64.78 | 32.12 | 41.10 | 77.07 | 55.10 |
| | IV (Ours) | **53.93** | **79.17** | 48.74 | **63.85** | **71.40** | 59.55 | **58.32** | 58.96 | **67.31** | **69.96** | 35.51 | **46.82** | **79.27** | **60.98** |

*Table 7.* Main experiment results with weighted-F1 as the metric. "Clean" denotes a standard one-shot ICL result.

| Dataset | 1-shot | | | 2-shot | | | 3-shot | | | 4-shot | | |
|---|---|---|---|---|---|---|---|---|---|---|---|---|
| | Clean | +IV | Diff | Clean | +IV | Diff | Clean | +IV | Diff | Clean | +IV | Diff |
| AG News | 86.96 | 88.17 | +1.21 | 87.99 | 89.04 | +1.05 | 87.87 | 88.84 | +0.97 | 89.01 | 89.32 | +0.31 |
| Rotten Tomatoes | 82.24 | 91.52 | +9.28 | 91.29 | 92.38 | +1.09 | 92.39 | 93.13 | +0.74 | 92.50 | 92.69 | +0.19 |

*Table 8.* Multi-shot clean and IV results using the Llama-2-70b model. The displayed metric is macro-F1. Conducted on 3 Nvidia RTX A6000 GPUs.

superior performance. We hypothesize that this result indicates a bias of FV towards the majority classes in this specific model. This bias results in an increased micro-F1 score; however, it causes the macro-F1 score to fall below the clean baseline, which is highly undesirable.

An additional experiment was conducted utilizing the Llama-2-70b model. Due to our computational budget constraints, it was not feasible to complete all tasks with a model of this scale. Therefore, we opted to conduct a multi-shot experiment, as described in Section 4.3 (Table 3), to more effectively showcase the efficacy of IV. The results are presented in Table 8.

## G. Comparison of Methodologies

We will begin with an introduction to the motivation and functioning of FV and TV. Following this, we will offer comprehensive comparisons from various perspectives.

**Function Vectors.** Function Vectors (Todd et al., 2023) are inspired by the observation that incorporating activations extracted from few-shot tasks on the last token at specific layers can prompt an LM to execute a task when applied to an unseen zero-shot prompt. To distill a more effective hidden-state representation, the researchers limit their investigation to attention heads. This decision is based on the heuristic that attention heads are the components used by transformers to transfer information across different token positions. The researchers aim to identify attention heads that have a causal influence on predicting the desired label for a given task. The method for calculating this causal effect is outlined as follows:

1. Compute the average activation $\bar{a}_{\ell j}^t$ of each attention head $j$ at layer $\ell$ over task $t$.

2. Feed the ICL prompt $\tilde{p}_i^t$ with shuffled labels into model $f$, and calculate the probability assigned to the target label $f(\tilde{p}_i^t)$.

3. Use one $\bar{a}_{\ell j}^t$ to replace the activation of its corresponding attention head, conducting a separate run for each instance. Subsequently, compute the edited probability for the target label again as $f(\tilde{p}_i^t | a_{\ell j} = \bar{a}_{\ell j}^t)$.

4. The *causal indirect effect* on task $t$ and the shuffled prompt $\tilde{p}_i^t$ is calculated as

$$\text{CIE}(a_{\ell j} \mid \tilde{p}_i^t) = f(\tilde{p}_i^t \mid a_{\ell j} := \bar{a}_{\ell j}^t) - f(\tilde{p}_i^t). \tag{27}$$

5. The *average indirect effect* is the average of the CIE across all tasks and prompts:

$$\text{AIE}(a_{\ell j}) = \frac{1}{|\mathcal{T}|} \sum_{t \in \mathcal{T}} \frac{1}{|\tilde{P}_t|} \sum_{\tilde{p}_i^t \in \tilde{P}_t} \text{CIE}(a_{\ell j} \mid \tilde{p}_i^t). \tag{28}$$

6. Gather the attention heads with highest AIE over all layers to serve as the activation source, forming set $\mathcal{A}$.

The researchers represent the contribution of $\mathcal{A}$ as a single vector by taking the sum of their average outputs, over a task, which is called a Function Vector for task $t$:

$$v_t = \sum_{a_{lj} \in \mathcal{A}} \bar{a}_{lj}^t. \tag{29}$$

To utilize FV, add it to the activation of the final token at a designated layer as the model processes a prompt.

One significant issue with FV is that it necessitates an extensive search through all attention heads of every layer, posing considerable scaling challenges as the model size grows. Theoretically, aside from the extraction time attributed to the extraction shot $k$, the extraction time of FV increases with an additional complexity of $O(E \times L \times H)$. Here, $E$ represents the number of extraction episodes, $L$ denotes the layer count of the LM, and $H$ is the number of attention heads in each layer. For example, GPT-J-6B has a total of 448 heads, while Llama-2-13B has 1600. This increase alone more than triples the time required to extract the FVs, not to mention the slower computation resulting from a longer prompt and a larger model size.

In contrast, Task Vector and our Iterative Vector do not encounter this issue and scale smoothly with larger models. During our experiments, we had to restrict the extraction shot $k$ for FV to maintain practical search times and ensure fairness across all evaluated methods, as mentioned in Section 4.

**Task Vectors.**    Task Vectors (Hendel et al., 2023) offer a mechanistic perspective on ICL. This approach conceptualizes ICL as a two-step process: first, a parameter vector $\theta$ is computed from the training sample, which is subsequently used to apply the "rule" defined by the vector to the query $x$.

There are many possible realizations of the above framework. The researchers presume that a simple way for a transformer to achieve this is for the initial $L$ layers to compute $\theta$. The remaining layers would take $\theta$ and $x$ as inputs to generate an output.

This provides a straightforward method to extract the language model's knowledge of a task and subsequently apply it. The process involves performing a forward pass of the transformer and utilizing the previously extracted $\theta$ to patch the $L$-th layer of the final token.

However, the boundary that separates this artificially divided two-stage process in the LM remains unclear and needs to be selected through empirical searching.

**Comparison with Iterative Vectors.**    The theoretical attributes of our methodology, in comparison to the baseline models, are as follows:

- FV and TV utilize their experiments to validate their respective hypotheses, rather than basing their methods on theoretical foundations.

- Consequently, their editing processes are heuristic and rely on intuition.

- Our proposed method is grounded in the meta-gradients derived from the demonstrations through the computation of the attention modules within the model.

- This approach not only identifies where to make edits (the attention layers) but also specifies how to perform the edits (by performing meta-gradient updates via adding to the activations).

The extraction and editing process differs considerably for each method, as illustrated below:

- FV examines all attention heads and aggregates the activations of the top-performing ones to obtain the vectors, which is highly time-consuming.

- TV simply identifies the optimal layer for the extraction and application of vectors.

- IV processes the activations from different classes separately, conducting aggregation and application based on this separation. We also propose iterative updates and batched extraction for meta-gradients, which have been proven to significantly enhance performance.

The hyperparameters specific to each method (instead of the evaluation framework) are as follows:

- FV: the count of top heads $|\mathcal{A}|$ and the layer to apply the vector.

- TV: the layer to apply the vector.

- IV: extraction strength $\alpha_1$, inference strength $\alpha_2$, and iterative batch size $b$.

Please refer to Appendix D for a more detailed discussion on the hyperparameters of IV.

As a side note, we can see from the comparisons above that there is considerable flexibility in the design of activation vectors. We hope that our efforts will serve as a catalyst for further exploration and advancement in this line of inquiry, ultimately unlocking the full potential of activation vectors.

## H. Concerning Zero-Shot Sequences

In both the FV and TV papers (Todd et al., 2023; Hendel et al., 2023), the vectors are utilized on zero-shot sequences. This aims to demonstrate the effectiveness of activation vectors in guiding the model as expected. The results confirm this: zero-shot sequences with activation vectors differ significantly from clean zero-shot runs. However, there remains a noticeable gap between zero-shot applications and standard few-shot ICL performance, which appears difficult to bridge. For instance, in Figure 4 of the TV paper, all FV runs fall behind the few-shot runs across all models, despite the tasks being simple synthetic ones.

Previous research has suggested reasons that may account for this disparity. Feng et al. (2023) provide fundamental impossibility results, indicating that language models cannot solve increasingly complex tasks in a single generation step. If we view the demonstration sequence as an extension of the inference steps generated by the LM—since the model treats all previous tokens equally, whether generated or provided—then without demonstrations, the LM's capabilities are significantly impaired. A zero-shot attempt might not provide adequate computation for the language model to address a given task. Consequently, it might be overly optimistic to expect activation vectors to circumvent all necessary computations.

Furthermore, Min et al. (2022b) demonstrated the importance of informing the LM about the label space of the current task to enhance ICL performance. In a zero-shot scenario, the model might struggle to focus its classification ability on the desired label, instead distributing it across the entire vocabulary space, as noted by Holtzman et al. (2021). This adds an extra burden for the model to extract meta-gradients and adjust accordingly.

Our early experiments on real-world tasks also confirmed that activation vectors do not perform well in a zero-shot setting. While there are some improvements, they remain inferior compared to the results achieved with even a one-shot approach. For synthetic experiments, these results may be adequate; however, to make activation vectors effective for practical applications, we must achieve better outcomes.

Table 2 of the FV paper offers an insight: FV is applied not only to zero-shot sequences but also to "uninformative" sequences, which are essentially few-shot sequences with shuffled labels. These shuffled sequences nearly double the performance compared to their zero-shot counterparts on synthetic tasks, prompting us to begin our investigation from this point. However, since using a shuffled sequence is not meaningful for our purposes, we employ a correct one-shot sequence instead. The advantages of this approach include a basic guarantee of performance, along with the presence of input-output separators in the support samples, which further facilitate the application of the vectors.

Taking all of the above into account, we have decided to focus on enhancing few-shot performance rather than zero-shot. Nonetheless, we hope our research will enhance future studies on activation vectors, enabling them to more effectively address the zero-shot scenario. This would represent a significant, albeit challenging, advancement.

---

**Algorithm 1** Extraction of Iterative Vectors

---

**Require:** extraction shot: $k$, extraction batch size: $b$, extraction strength: $\alpha_1$
**Ensure:** extracted Iterative Vector: $\mathbb{V}$

  1: $\mathbb{V} \leftarrow \varnothing$                                             ▷ Initialize the variable to store the IV
  2: IVs $\leftarrow \varnothing$                                ▷ An empty list to store IV for each episode
  3: **for** the $i$-th episode **do**
  4:      Support, Query $\leftarrow$ RANDOMEPISODE($k$)                 ▷ Sample a $k$-shot episode
  5:      Order, Support $\leftarrow$ SHUFFLE(Support)            ▷ Shuffle and remember the classes
  6:      SupportAndQuerySequence $\leftarrow$ VERBALIZE(Support $\oplus$ Query)     ▷ Convert to few-shot prompt
  7:      QuerySequence $\leftarrow$ VERBALIZE(Query)             ▷ Convert to zero-shot prompt
  8:      SupportAndQueryVector $\leftarrow$ APPLYANDEXTRACT(SupportAndQuerySequence, $\mathbb{V}$, $\alpha_1$)
  9:      QueryVector $\leftarrow$ EXTRACT(QuerySequence)       ▷ See Algorithm 3 for the episodic functions
10:      **for** each class of the task **do**
11:          $p \leftarrow$ the position(s) where order is equal to class         ▷ Collect by each class
12:          $v$[class] $\leftarrow$ MEAN(SupportAndQueryVector[$p$] $-$ QueryVector)     ▷ Average over shots
13:      **end for**
14:      $v$[QUERY] $\leftarrow$ SupportAndQueryVector[QUERY] $-$ QueryVector      ▷ Collect the query as well
15:      IVs $\leftarrow$ IVs $\cup \{v\}$               ▷ Append the current episode's IV to the list
16:      **if** $i \bmod b = 0$ **then**          ▷ Check if the current episode is a multiple of batch size
17:          $\mathbb{V} \leftarrow$ MEAN(IVs)         ▷ Update the IV to apply as the average over episodes
18:      **end if**
19: **end for**

---

**Algorithm 2** Evaluation

---

**Require:** evaluation shot $k'$, extracted Iterative Vector: $\mathbb{V}$, inference strength: $\alpha_2$
**Ensure:** classification labels: Results

  1: Results $\leftarrow \varnothing$               ▷ An empty list to store results for each episode
  2: **for** the $i$-th episode **do**
  3:      Support, Query $\leftarrow$ RANDOMEPISODE($k'$)        ▷ Sample an episode, typically with $k' = 1$
  4:      Support $\leftarrow$ SHUFFLE(Support)       ▷ Shuffle to avoid patterned few-shot sequence
  5:      SupportAndQuerySequence $\leftarrow$ VERBALIZE(Support $\oplus$ Query)         ▷ Convert to prompt
  6:      Logits $\leftarrow$ APPLY(SupportAndQuerySequence, $\mathbb{V}$, $\alpha_2$)         ▷ Run the LM with editing
  7:      Results $\leftarrow$ Results $\cup \{$ARGMAX(Logits)$\}$       ▷ Calculate the classification result
  8: **end for**

---

---

**Algorithm 3** Episodic Functions

---

1: **function** EXTRACT(sequence) ▷ Extracts activations from the LM
2:   $v \leftarrow \varnothing$
3:   **run** LM(sequence) **with** ▷ Hook into the LM with the following operations
4:    **for** each layer in LM **do** ▷
5:     $p \leftarrow$ the position of the input-output separator after the query
6:     $v \leftarrow v \cup \{\text{Attn}[p]\}$ ▷ Store the activation of each attention layer
7:    **end for**
8:   **end run**
9:   **return** $v$
10: **end function**

11: **function** APPLY(sequence, $\mathbb{V}$, $\alpha$) ▷ Apply IV to LM inference process
12:   **run** logits $\leftarrow$ LM(sequence) **with**
13:    **for** each layer in LM **do**
14:     **for** each support sample in sequence **do**
15:      $p \leftarrow$ the position of the input-output separator after the sample
16:      $c \leftarrow$ the class of the sample
17:      $\text{Attn}[p] \leftarrow \text{Attn}[p] + \alpha \times \mathbb{V}[c]$ ▷ Edit the separators in the support sequence...
18:     **end for**
19:     $p \leftarrow$ the position of the input-output separator after the query
20:     $\text{Attn}[p] \leftarrow \text{Attn}[p] + \alpha \times \mathbb{V}[\text{QUERY}]$ ▷ ...as well as the query
21:    **end for**
22:   **end run**
23:   **return** logits
24: **end function**

25: **function** APPLYANDEXTRACT(sequence, $\mathbb{V}$, $\alpha$) ▷ Apply the IV during extraction
26:   $v \leftarrow \varnothing$
27:   **run** LM(sequence) **with**
28:    **for** each layer in LM **do**
29:     **if** $\mathbb{V} \neq \varnothing$ **then** ▷ The first batch does not have $\mathbb{V}$ for editing
30:      **for** each support sample in sequence **do**
31:       $p \leftarrow$ the position of the input-output separator after the sample
32:       $c \leftarrow$ the class of the sample
33:       $\text{Attn}[p] \leftarrow \text{Attn}[p] + \alpha \times \mathbb{V}[c]$ ▷ Edit (support)
34:      **end for**
35:      $p \leftarrow$ the position of the input-output separator after the query
36:      $\text{Attn}[p] \leftarrow \text{Attn}[p] + \alpha \times \mathbb{V}[\text{QUERY}]$ ▷ Edit (query)
37:     **end if**
38:     $p \leftarrow$ the position of the input-output separator after the query
39:     $v \leftarrow v \cup \{\text{Attn}[p]\}$ ▷ Extract and append to list
40:    **end for**
41:   **end run**
42:   **return** $v$
43: **end function**

---

