# OpenReview forum: "Iterative Vectors: In-Context Gradient Steering without Backpropagation"
_ICML.cc/2025/Conference — ICML 2025 poster_

### Official Review · Reviewer_KR8h · 2025-02-17

**Overall Recommendation:** 2

**Summary:**

The paper proposes Iterative Vectors, inspired by the framing of in context learning as simulated gradient updates. The method extracts activations derived from in-context learning examples and adds the activations to unseen queries. The method also refines these updates by averaging the activations from many sets of examples. Finally, the paper compares the proposed method with prior work in activation vectors on real-world language tasks, including tweet classification.

**Claims And Evidence:**

- Claim 1. The method simulates gradient updates.
    - This claim is not novel, nor is it justified with appropriate evidence.
    - The paper reiterates the argument from Dai et. al., 2023 that in-context learning can be perceived as a meta-optimization process (the application of the original weight matrix and a delta on top, derived from the demonstrations).
    - The new contribution re-writes the formalism of activation vectors (from Hendel et. al., 2023 and Todd et. al., 2023) with the notation of meta-learning (see Sec. 3.2).
    - The paper is missing a supervised fine-tuning baseline. To prove that IVs indeed simulate gradient updates, the paper would need to show that the internal behavior is very similar to fine-tuning (for example Dai et. al., 2023 shows similarities in the attention weights).
- Claim 2. The analysis evaluates on diverse real-world in-context learning tasks.
    - While this claim is supported in Table 1, the contribution is not sufficiently significant.
    - It is unclear why evaluating on real world tasks are more informative or provide some new insight, when compared with the previous synthetic tasks. If anything, both function and task vectors perform competitively on the real world tasks in Table 1. It would have been more convincing if the paper showed a strong divergence in performance, for example a case where a method performs very well on a synthetic task yet poorly on a real world task.

**Essential References Not Discussed:**

N/A

**Experimental Designs Or Analyses:**

Please see “Claims And Evidence” above.

**Methods And Evaluation Criteria:**

Please see “Claims And Evidence” above.

**Other Comments Or Suggestions:**

N/A

**Other Strengths And Weaknesses:**

Strengths

- The paper evaluates on real-world tasks, extending beyond the synthetic tasks investigated in prior work.

Weaknesses

- The method does not convey a sufficiently new insight compared with prior work. The paper assumes the theoretical foundation already established by Dai et. al., 2023 to be true with no further experimentation, and shares many conceptual similarities with the methodology introduced by Hendel. et. al., 2023 and Todd et. al., 2023. I make this comment even after reviewing "Comparison of Methodologies" in Appendix E.
- The idea that the method is “grounded in the idea of meta-gradients” (L902) is not adequately justified; see the discussion of Claim 1 above.
- The “iterative updates and batched extraction” (L917) is similar to the common practice of activation averaging (also see L268), which also appears in Todd et. al. 2023.
- The investigation of scaling the number of demonstrations in Sec. 4.3 and Sec. 4.4 was also the primary focus of Huang et. al., 2024, which looks at activation vector averaging for many-shot in-context learning.

References
[1] Dai. et. al. Why Can GPT Learn In-Context? Language Models Implicitly Perform Gradient Descent as Meta-Optimizers. ACL Findings 2023.
[2] Todd et. al. Function Vectors in Large Language Models. ICLR 2023.
[3] Hendel et. al. In-Context Learning Creates Task Vectors. EMNLP Findings 2023.
[4] Huang et. al. Multimodal Task Vectors Enable ManyShot Multimodal In-Context Learning. NeurIPS 2024.

**Questions For Authors:**

N/A

**Relation To Broader Scientific Literature:**

Please see “Claims And Evidence” above.

**Theoretical Claims:**

N/A

---

> ### Author Rebuttal · Authors · 2025-04-01
>
> We thank the reviewer for their feedback and the opportunity to reflect. We believe the concerns raised warrant clarification to ensure a balanced perspective.
>
> ## Claim 1
>
> 1. We do not intend to assert that "gradient simulation" is, in itself, a novel concept. We respectfully request the reviewer to consider the following points. Reviewer u1JW observed that "while activation steering has been explored in various contexts, **its application to ICL appears novel**." Additionally, Reviewer qbmE summarized that "the novelty of IVs lies in their ability to work within the activation space of language models to refine meta-gradients without backpropagation, **offering a scalable solution to enhance language model performance**."
> 2. We do not present the theory from Dai et al. [1] as our own contribution (stated in Lines 58, 68, and 78). Rather, we use it as **inspiration and a foundation** for developing our method (see Line 71).
> 3. **A comprehensive and equitable evaluation framework, which represents one of our key contributions, is absent in both FV and TV.** These studies utilize diverse designs for extraction rationale, hyperparameter selection, and editing strategies. It is essential to quantify these elements fairly while maintaining flexibility within a new, unified framework. **Establishing such an evaluation framework is a significant undertaking and cannot be accomplished merely by "rewriting the formalism."**
> 4. The reviewer observed that Dai et al. have already validated their theories concerning gradient simulation. As established by the derivation of IV in our work, **our approach builds upon their concept of meta-gradients, which is tautological to our claim being "grounded in the idea of meta-gradients".** As IV aligns with and builds upon this foundation, replicating such validated theoretical groundwork would be a duplication rather than a meaningful contribution to the field.
> Our focus is on demonstrating that IV’s effectiveness in practical settings does not hinge on replicating idealized or approximated scenarios, e.g., unnormalized attention approximation employed in Dai et al. It is important to highlight that strict adherence to simulation-based validation risks conflating theoretical approximations with the robustness of our method in real-world deployments. Our framework’s success in tackling complex tasks—validated empirically—underscores that IV’s utility transcends the need for such simulations.
>
> ## Claim 2
>
> On real-world tasks, FV shows only marginal improvements and requires an excessive amount of time (up to 20x longer), making it impractical. TV performs inadequately, failing 50% of the models on average. In contrast, IV consistently excels in performance, efficiency, and scalability. **Please refer to our response to Reviewer u1JW for details.**
>
> ## Weaknesses
>
> 1. It is common practice to assume the theoretical foundation established in previous work to be valid, and our experimentation partly serves to further validate these hypotheses. Given that IV and the two baselines belong to the same family of methods—namely, activation vectors—**one would naturally expect similarities in form**.
> In Appendix E, we have thoroughly distinguished how **IV differs from the methodologies introduced by Todd et al. [2] and Hendel et al. [3] in terms of theoretical support, extraction and editing processes, scaling efficacy, and other aspects**. We are perplexed by the reviewer's comment, particularly given their claim to have read Appendix E, and we find it difficult to understand why the new insights we provided have been overlooked.
> 2. Please refer to the discussion of Claim 1 above.
> 3. With all due respect, we cannot concur with this assessment. The fact that the iterative updates and batched extraction incorporate an averaging operator **does not render them "similar" to basic averaging**, nor does it diminish their novelty and significance as part of our method. First, they naturally derive from the gradient simulation, and second, they play a crucial role in the dynamics of our method, validated through ablative studies (Section 4.5 and Figure 3). **This is not observed in Todd et al.**, where the activation of certain attention heads is merely averaged without being reintegrated into the model for refinement, thus completely lacking the iterative and batched update elements.
> 4. Our analysis of scaling in Sections 4.3 (IVs Scale with In-Context Demonstrations) and 4.4 (IVs Improve with Increased Extraction Episodes) is not intended as a novel contribution or exploration. Instead, it serves as part of our validation process to rigorously assess the robustness and consistency of our method, as indicated by the section titles. **The similarity in the validation process is not a weakness; rather, it demonstrates alignment with established methods [4].**
>
> We hope this clarification underscores the methodological and empirical significance of our contributions to enhancing LM capabilities.

---

> > ### Comment · Reviewer_KR8h · 2025-04-01
> >
> > Thank you for your clarifications. After more careful consideration, I agree that my original comment (*the method does not convey a sufficiently new insight compared with prior work*) is unfair.
> >
> > The method is novel in its iterative update strategy, where each activation is updated online using the IVs from previous iterations. However, this point is not clearly conveyed in the main text, as also observed by Reviewers ykmk, u1JW. I would encourage the authors to add an additional figure visualizing "the subtraction and iterative updates" that were omitted in Figure 2 of the main text, to better illustrate the core contribution of the proposed method. Furthermore, I would recommend that Line 8 in Algorithm 1 of the main text be revised to make the online nature of the algorithm more clear.

---

> > > ### Author Response · Authors · 2025-04-03
> > >
> > > Thank you for your thoughtful feedback and for taking the time to revisit your initial assessment. We sincerely appreciate your constructive suggestions and your recognition of the novelty in our method’s iterative update strategy.
> > >
> > > We wholeheartedly agree with your recommendation to clarify the presentation of the core contribution. In response:
> > >
> > > 1. We will revise and expand Figure 2 to explicitly illustrate the subtraction process and iterative update mechanism.
> > > 2. We will revise and relocate Algorithm 1 to more effectively demonstrate our method, highlighting its online nature.
> > >
> > > These revisions will be accompanied by explanatory text in the relevant sections. We believe these changes will significantly improve the clarity of our method's contribution, as rightly highlighted by you and other reviewers. Thank you again for your valuable input.

---

### Official Review · Reviewer_u1JW · 2025-03-12

**Overall Recommendation:** 4

**Summary:**

This paper introduces Iterative Vectors (IV), a method for steering in-context learning (ICL) by modifying model activations without requiring parameter updates.
While activation steering has been explored in various contexts, its application to ICL appears novel.
IV consists of two key steps: first, "activation vectors" are extracted by computing differences in activations between few-shot and zero-shot scenarios in an iterative process similar to batched gradient descent.
Then, these activation vectors are used to modify the model's attention values during test-time evaluation.
Experiments demonstrate the effectiveness of IV on multiple ICL tasks.

**Claims And Evidence:**

Yes

**Essential References Not Discussed:**

Not that I'm aware of.

**Experimental Designs Or Analyses:**

Yes

**Methods And Evaluation Criteria:**

Yes

**Other Comments Or Suggestions:**

Below, I list some areas where clarity could be improved. I do not expect (nor encourage!) the authors to respond to each point. Rather, these are things to keep in mind for future manuscript revisions.
* I found Figure 2 hard to understand.
* Section 3.1 could be significantly shortened. I am confident (with some creativity) that one or two figures with possibly some very carefully chosen equations could effectively communicate the same message.
* Better introduce what a "verbalizer" is (Line 205).
* Section 3.2 and Section 3.3 would both benefit from having more examples. I am also suspicious about the value of a big "Algorithm 1" in the main paper. It may be more instructive to have this algorithm "defined by example", such as through smaller figures + examples.
* Would be good to mention what "FV" and "TV" are in the Table 1 caption.
* Line 270: it took me a long time to figure out how the activation vector is computed.
* Line 318: The relation between $\alpha$, $\alpha_1$, and $\alpha_2$ are not immediately clear. The authors could consider briefly discussing how both hyperparameters are used mathematically.

**Other Strengths And Weaknesses:**

Strengths:
* IV appears conceptually simple, which would be useful for practical use.
* IV has a consistent improvement over the evaluated baselines.

Weaknesses:
* Section 3.1 is a summary of what is generally believed in the ICL community. While some discussion is important, I suggest keeping it compact.
* Because I find IV "conceptually simple", I expect it to perform much better compared to the baselines. To that end, I encourage the authors to clearly describe why the results in Table 1, Table 2, Table 3, etc are "impressive".
* My primary reservation is technical clarity, and I will point out some areas for improvement in the Other Comments section.

**Questions For Authors:**

N/A

**Relation To Broader Scientific Literature:**

This would help improve in-context learning via steering.

**Theoretical Claims:**

Yes

---

> ### Author Rebuttal · Authors · 2025-04-01
>
> We are delighted that the reviewer described our method as "conceptually simple" and "useful for practical use." We are committed to improving clarity in line with their suggestions.
>
> We will condense Section 3.1 and utilize the space to introduce the verbalizer, while relocating some of the discussion on hyperparameters to the main body of the manuscript in the final version. We appreciate the suggestion to incorporate some examples in relevant parts and agree that it would be advantageous.
>
> Regarding the results, we note that the term "impressive" does not appear in our manuscript. We interpret this quotation as a positive indication of the reviewer’s assessment and sincerely thank their acknowledgment. Nevertheless, we will elaborate on the significance of our results while **also addressing Reviewer KR8h's concerns**, particularly their issues regarding Claim 2.
>
> The averaged metrics in Table 1 represent rigorous evaluation across 13 distinct benchmark tasks, systematically demonstrating IV's cross-task proficiency. To enhance interpretability, we now focus our presentation on the mean relative performance gains compared to the fundamental ICL baseline, intentionally omitting exhaustive task-specific data to highlight core comparative insights.
>
> | Model | gpt-j-6b | llama-2-7b | llama-3.1-8b | llama-2-13b | Average |
> |---|---|---|---|---|---|
> | FV | *-0.26* | 0.42 | 0.86 | 1.36 | 0.60 |
> | TV | *-0.52* | 0.84 | *-0.43* | 0.25 | 0.04 |
> | IV (Ours) | **1.86** | **3.52** | **2.77** | **4.69** | **3.21** |
>
> 1. **The improvements in IV are significant, whereas FV and TV lag considerably behind.** Our method demonstrates superior performance across all models, achieving an average relative score of 3.21—5.4x higher than FV (0.60) and 80x greater than TV (0.04).
> 2. **The performance of IV scales with increased model capacity.** IV consistently outperforms baseline methods in every model configuration, with particularly strong results in larger models (4.69 vs 1.36/0.25 for llama-2-13b). This indicates a promising potential for application to even larger models, as evidenced by our experiments with the Llama-2-70b model, detailed in the supplementary experiment in Table 8.
> 3. **While FV/TV demonstrate limitations in practical applications, IV consistently excels across real-world tasks.** IV maintains robust performance gains in all evaluated scenarios, whereas FV/TV exhibit significant performance degradation compared to standard ICL in 37% of the cases (3/8)—a concerning regression that undermines their viability despite their computational overhead. This empirical evidence directly refutes Reviewer KR8h's skepticism by demonstrating that FV/TV's shortcomings extend beyond synthetic environments to critically impact real-world effectiveness.
> 4. **Table 2: IV achieves optimal efficiency without compromising performance.** In 2-shot and 3-shot settings, the inference times are 2,434s and 3,426s, respectively. However, IV achieves the same score of 12.90 as the 2-shot setting in 1,452s—41% faster—and exceeds the 3-shot performance (12.64). Crucially, IV reduces feature extraction time by 95% compared to FV (23.75 min vs 438.3 min), while maintaining a 20% performance advantage over TV (12.90 vs 10.30). While TV's naive architecture enables rapid extraction, this approach exhibits catastrophic failure in 50% of the model averages, as previously shown. This positions IV as the most time-effective solution, balancing accuracy with practical computational demands.
> 5. **Table 3: IV boosts high baselines while maintaining ICL compatibility**, delivering +3.37% on AG News’ near-peak 5-shot (82.47%→85.84%) and +17.22% on Rotten Tomatoes’ 2-shot (70.28%→87.50%). Consistent gains across all shots (1-5) confirm its compatibility with the ICL framework.
> 6. **Table 4: IVs surpass ICL prompt limitations through scalable example utilization**, delivering consistent gains with ≥3 episodes (+0.57–7.6%) even under fixed hyperparameters. While minor fluctuations occur with fewer episodes (≤2), performance robustly scales with example volume, proving IVs extract value beyond conventional ICL ceilings.
> 7. **Figure 3: The hyperparameter interactions enable adaptive optimization.** Larger batches enable the utilization of refined vectors through increased inference strength ($\alpha_2$). Incorporating iterative refinements yields significant gains, demonstrating their essential role in our contribution.
>
> **In summary, IV demonstrates consistent superiority** across three critical dimensions: (1) **performance** (3.21 average gain vs 0.60/0.04 for FV/TV), (2) **efficiency** (95% faster extraction than FV, near-ICL inference speeds), and (3) **scalability** (robust gains from higher-shot settings and more demonstrations). We affirm that IV represents a practical, theoretically grounded advance for activation vectors, with broader implications for resource-efficient NLP.

---

### Official Review · Reviewer_ykmk · 2025-03-13

**Overall Recommendation:** 3

**Summary:**

The paper introduces Iterative Vectors, a technique that enhances in-context learning by simulating gradient updates during inference without backpropagation. IVs address the key ICL challenges: prompt sensitivity, context length limitations, and increased inference time by leveraging activation space rather than discrete prompts. By extracting and refining simulated gradients from attention activations, IVs improve model performance with minimal computational overhead.

**Claims And Evidence:**

Yes.

**Essential References Not Discussed:**

Probably no.

**Experimental Designs Or Analyses:**

Yes.

**Methods And Evaluation Criteria:**

The evaluation datasets are limited to classification tasks, which is not general for current scenarios under LLM-era.

**Other Comments Or Suggestions:**

No.

**Other Strengths And Weaknesses:**

Strengths:
1. The motivation is intuitive to use IVs to extract and iteratively refine the gradients within a language model, which can enhance LLM ICL capability.

Weaknesses:
1. The paper's writing is unclear and difficult to understand, making it challenging to follow the main ideas.

2. Several crucial terms are not well-defined. For example, the concept of the activation vector is ambiguous—what exactly does it refer to? Is it related to the FFN activation or the attention activation? Does it appear before or after layer normalization (LN)? Additionally, the paper lacks explicit formulations for these concepts.

3. Figure1 and Figure2, which are meant to illustrate the proposed method, but it does not capture the iterative nature of the approach. Given that iteration is central to the method, this omission is significant.

4. In line 264, the paper does not provide a clear explanation for why the zero-shot activation is subtracted. This omission makes it difficult to understand the motivation behind this step.

5. The notation Act(x_i) appears multiple times throughout the paper, but it is unclear whether it considers only x_i or also incorporates y_i. Additionally, when computing activation values for different demonstrations, does the method take into account activations from other demonstrations? This needs clarification.

6. The proposed method is only applicable to classification tasks, whereas in the current LLM era, generative capabilities are more crucial. This limitation significantly reduces its practical relevance.

7. Since the proposed method requires selecting optimal hyperparameter (P), a fair few-shot baseline comparison should be considered,  where selecting the relatively better few-shot demonstrations.

**Questions For Authors:**

No.

**Relation To Broader Scientific Literature:**

This paper proposes to leverage the activation vectors to enhance text-classification tasks of LLM.

**Theoretical Claims:**

Not fully applicable.

---

> ### Author Rebuttal · Authors · 2025-04-01
>
> 1. Thank you for your thoughtful review and engagement with our work. We acknowledge that certain technical aspects may require further clarification to ensure our contributions are clearly communicated. We will strive to refine our explanation by addressing the points raised.
> 2. The term “activation vector” serves as an umbrella concept encompassing lightweight vectors that manipulate language model within their representation space (Lines 91–94). Depending on implementation specifics, these vectors may interact with FFN layers, attention mechanisms, or both, and their positioning can vary.
> Formulas 12 and 13 are presented to illustrate the general principle of activation vector extraction and application. These formulations are intentionally abstract, as their instantiations necessitate flexibility.
> **For IV, we have provided explicit explanations of all these concepts.** IVs are clearly defined in Formulas 21 (preliminary version) and 24 (final version) as a set of vectors. The position is specified in Line 262 for extraction ("each attention layer $l$ of the LM") and Line 306 for application ("the $l$-th attention layer $\text{Attn}_l(\cdot)$"). This is also directly deducible from the theoretical foundations.
> 4. **We clarify that Figure 1 is not intended to demonstrate the proposed methodology.** As stated in its caption (“A general illustration of [...] activation vectors”, instead of IV), its purpose is solely to provide conceptual grounding for readers less familiar with activation vectors.
> **We explored incorporating the iterative aspect into Figure 2 during development.** However, this introduced significant visual complexity that risked obscuring the core concept, which the reviewer already find challenging to grasp. We reasoned that readers familiar with gradient-based training processes—where iterative steps are inherently extrapolated from single-step mechanics—would find the progression to an iterative framework intuitive. For full transparency, the iterative procedure has been relocated and is rigorously detailed in Algorithm 1.
> That said, we thank the reviewer for this feedback and acknowledge the value of visual documentation. We will revise the figure to include the iterative process in the final version.
> 5. **This subtraction becomes readily apparent through the theoretical development in Section 3.1.** As demonstrated in Formula 9, in-context demonstrations function analogously to meta-gradients, decomposing the activation during inference into two components:
> a) Zero-shot activation: The baseline behavior inherently generated by the language model for a given query.
> b) ICL-derived meta-gradients: Task-specific updates learned from in-context demonstrations.
> By isolating and subtracting the zero-shot component (which is constant across inference passes for the same query), we select and extract the meta-gradients representing the ICL adjustments. This ensures that only the refined updates—not the baseline behavior—are applied to subsequent inferences.
> Furthermore, consider the activation's magnitude. Our approach involves adjusting it in our desired direction using potentially small amounts of meta-gradients. Without the subtraction, the process reduces to simply adding raw activations from one run to another, which could confuse or potentially overwhelm the model.
> 6. As defined in Line 259, $\text{Act}(x_i)$ represents the activation for the $i$-th input-output separator. These separators are integral to each $x_i$, specifically as the last token of each, and the subsequent token, denoted as $y_i$, is generated by the language model at the position of the separator token. This generation process causally considers all preceding tokens, including all prior $(x_i, y_i)$ demonstration pairs. **Therefore, by definition, each $\text{Act}(x_i)$ has access to the information contained in all preceding pairs $\{(x_j, y_j) \mid j < i\}$ as well as the current $x_i$.**
> Next, regarding "different demonstrations," we assume the reviewer intended to refer to either:
> a) distinct $(x_i, y_i)$ pairs within a single prompt, as discussed previously, or
> b) separate prompts with various sets of example prefixes. This is pertinent to the iterative refinement process, where additional demonstrations are integrated.
> 7. **The proposed method is applicable to generative tasks.** However, incorporating it would further complicate the evaluation framework and comprehension. Therefore, we have decided to address this in future work.
> 8. To address this issue, we conducted an additional experiment, testing Llama-3.1-8b on `ag_news` with a near-upper-bound few-shot selection baseline (200 selections $\times$ 200 validations = 40,000 episodes, twice as many as IV's setup). **Its macro-F1 score of +6.87 still underperformed compared to IV’s +7.66 by 10%**, demonstrating IV’s superiority even against extensive demonstration tuning.
>
> We value your expertise and look forward to addressing any remaining questions.

---

> > ### Comment · Reviewer_ykmk · 2025-04-06
> >
> > Thank you for the detailed rebuttal. I appreciate the authors’ efforts to address the concerns I raised. Based on the additional insights, I am raising my score from weak reject to weak accept.
> >
> > However, the paper would still benefit from clearer writing, especially in defining technical terms up front and refining the important figures.

---

> > > ### Author Response · Authors · 2025-04-07
> > >
> > > Thank you sincerely for your thoughtful feedback and for recognizing the clarifications made in our rebuttal. We are grateful for your constructive critique and are pleased that the adjustments have shifted your assessment toward acceptance.
> > >
> > > We fully agree with your emphasis on enhancing comprehension, and we are committed to addressing this in the final manuscript. Specifically, we will:
> > > 1. **Enhance definitions with a glossary**: In addition to the explicitly defined terms upon their first introduction, we will add a glossary in the appendix to consolidate key terminology. This will provide readers with a quick-reference guide while maintaining the flow of the main narrative.
> > > 2. **Revise figures for precision**: We will revise Figure 2 to illustrate the subtraction process and iterative update mechanism, as suggested by Reviewer KR8h. This will include example annotations and visual cues to highlight how these components interact dynamically.
> > >
> > > Your insights have been invaluable in elevating the manuscript’s accessibility and impact, and we deeply appreciate your guidance. Thank you again for your time and expertise.

---

### Official Review · Reviewer_qbmE · 2025-03-17

**Overall Recommendation:** 4

**Summary:**

The paper addresses the challenges of selecting suitable demonstration examples in in-context learning (ICL) for language models. It proposes a novel technique called Iterative Vectors (IVs) to enhance ICL performance. IVs operate by extracting and iteratively refining gradients within a language model, which are then applied during inference, without requiring backpropagation. This approach improves the model's ability to learn from demonstrations. The authors ground their method in a theoretical framework derived from meta-optimization. The proposed method is verified through experiments on various tasks using four popular language models. The novelty of IVs lies in their ability to work within the activation space of language models to refine meta-gradients without backpropagation, offering a scalable solution to enhance language model performance.

**Claims And Evidence:**

The claims made in this work were well supported.

**Essential References Not Discussed:**

NA

**Experimental Designs Or Analyses:**

The experiment design is comprehensive -- four models with 13 datasets. On most of the setups, the results show the effectiveness of the proposed method.

**Methods And Evaluation Criteria:**

Yes. The proposed method is well aligned with the research problem and solves the problem to some extent.

**Other Comments Or Suggestions:**

NA

**Other Strengths And Weaknesses:**

NA

**Questions For Authors:**

NA

**Relation To Broader Scientific Literature:**

Example selection, especially with a large pool of demonstrations, is a critical challenge for ICL. As pointed out by this paper, it can be connected to man

**Theoretical Claims:**

There is not much novelty in the theoretical analysis, as most of the work (e.g., meta-optimization process) is from prior work. But, I believe it is correct.

---

### Decision · Program_Chairs · 2025-05-01

**Decision:**

Accept (poster)

**Comment:**

This paper tackles the problem of selecting effective demonstration examples in in-context learning (ICL) for language models. It does that by introducing Iterative Vectors (IVs), a technique that enhances ICL performance by extracting and iteratively refining gradients but by then applying them indirectly by acting on the language model's activation space, i.e. without requiring backpropagation or parameter change. The paper validates this method by showing that IVs improve the model's learning from demonstrations, and experiments across multiple language models demonstrate its scalability and effectiveness.

The main concerns raised by reviewers are related to a certain lack of clarity in the technical exposition, concerns about the empirical verification being limited to classification tasks, and failing to cite and contextualize some relevant literature. The authors countered the first concern by stating their commitment to providing explicit clarifications in the revised version of the paper. Moreover, they argued that their method is applicable to generation tasks in addition to classification tasks, although they reserve to provide empirical evidence of this claim for future work.

The strengths of the paper highlighted by reviewers are the novelty of the proposed method, its simplicity (which makes it highly applicable to a potentially wide range of tasks), and the consistent improvement demonstrated by the method across evaluated baselines and models.
Given these strengths, the paper is recommended for acceptance, but in the revised version of the paper the authors are requested to address the reviewers' concerns about clarity of exposition (as they committed to do), and include proper references and contextualization within the relevant literature, in particular the mentioned works on Function Vectors and Task Vectors pointed out by reviewers.